# Young Adults Rehabilitation experiences and Needs following Stroke (YARNS): A scoping review of the rehabilitation care experiences and outcomes of young adults post-stroke

Colin Chandler[1☯]*, Catherine Clarissa[1,2☯], Arcellia Farosyah Putri[3☯], Lissette Aviles[1☯], HyeRi Choi[4☯], Jonathan Hewitt[5‡], Emily Hennessy[1‡], Helen Noble[6‡], Joanne Reid[6‡], Aisha Holloway[1‡], Daniel Kelly[7‡]

1 Nursing Studies, University of Edinburgh, Scotland, United Kingdom, 2 Scottish Collaboration for Public Health Research and Policy (SCPHRP), University of Edinburgh, Scotland, United Kingdom, 3 Dr Cipto Mangunkusumo Hospital, Jakarta, Indonesia, 4 School of Nursing, Hong Kong University, Hong Kong, China, 5 Department of Population Medicine, Cardiff University, Wales, United Kingdom, 6 School of Nursing and Midwifery, Queen's University, Belfast, United Kingdom, 7 Department of Healthcare Sciences, Cardiff University, Wales, United Kingdom

☯ These authors contributed equally to this work.
‡ These authors also contributed equally to this work.
* colin.chandler@ed.ac.uk

## Abstract

### Background

Stroke is one of the leading disabling conditions in the United Kingdom. While an increasing focus on the rehabilitation care for stroke survivors has been noted, practice guidelines, targets and services tend to be tailored for the elderly population rather than young adults.

### Objectives

This scoping review aimed to map the existing literature on the rehabilitation care experiences and outcomes of young adults post-stroke aged between 18 and 45 years within acute and social care settings.

### Methods and analysis

A scoping review was conducted to identify existing practice and published academic and evidence-based literature related to the topic. Arksey and O'Malley's framework guided the review and reporting followed the PRISMA-ScR checklist. Electronic databases and grey literature were searched to capture the potentially recent and relevant literature available in English from year 2000–2022. The databases included CINAHL, MEDLINE, EMBASE, PsycINFO, ASSIA, AMED, PEDro, Cochrane Library and Web of Science. Grey literature was searched from the following sources: Google Scholar, websites of networks, organisations and charities related to stroke. Hand searches were performed on the reference lists of the included articles and grey literature to find relevant references. Data were extracted on young adults' experiences of stroke and rehabilitation care and their outcomes and experiences of a particular stroke rehabilitation programme in acute and social care settings.

**Data Availability Statement:** As this is a review of published literature, all relevant data are available from the sources cited in text and included in the reference list.

**Funding:** This work was supported by the SameYou Charity through an RCN Foundation grant. The funders had no role in study design, data collection and analysis, decision to publish, or preparation of the manuscript.

## Results

Eighty-five articles were included in this review. In most instances, stroke was reported to negatively impact young adults, although positive consequences were also documented. The focus and the outcomes of stroke rehabilitation were predominantly physical especially in the areas of movement, communication and memory. Young adults experienced a lack of age-adapted stroke rehabilitation in acute and social care settings.

## Conclusion

Our results highlight the unmet needs of young adults in their stroke recovery journey. Effective rehabilitation programmes and interventions should be developed to support young adults following stroke and meet their age-specific needs.

## Introduction

Stroke is the third leading cause of disability worldwide and number one among neurological disorders [1,2]. In the UK context, stroke is one of the foremost disabling conditions affecting the population [3]. Although the risk of stroke increases significantly with age [4], in the UK around 6% of new strokes are in the 18–45 age group and almost half of long term stroke survivors are aged between 20–64 years [5].

Stroke in young adults may result in sudden death or a life with severe, moderate, mild or no apparent disability [6]. Post-stroke challenges can include physical, emotional and psycho-social aspects that change younger adults' lives significantly [7,8]. These changes also impact on how they return to a variety of social activities after stroke such as family, childbearing or childrearing and work [9]. The stroke recovery journey of young adults is unique to each individual, as such this may involve reshaping their life goals and how they perceive themselves [10].

Given their unique recovery journeys and their age-specific needs around family, relationships and career or employment, stroke rehabilitation for young adults should be tailored to the individual [10]. However, research on stroke rehabilitation often assimilates people who have had a stroke as a homogenous group regardless of age. Guidelines, targets and services tend to be focused more on the elderly population and may not address the needs of younger adults following stroke [11,12]. Therefore, there is a pressing need to explore the experiences of young adults diagnosed with stroke, the scale of the needs, and the availability of rehabilitation services to support them. This scoping review aimed to explore the existing literature on the impact of stroke on young adults, the focus and the outcomes of the rehabilitation programmes available to them, and their experiences of accessing stroke rehabilitation across acute and community settings. In the literature the definition of young adults ranges from 18–45 [13–15], 18–55 [8,16,17] and 18–65 years [7,18]. For the purposes of this review we defined young adults as aged between 18 to 45 years.

## Methods

The scoping review approach was used to capture and to summarise the body of literature on young adults' experiences of stroke and stroke rehabilitation from a broad perspective and in a range of settings. The review was structured on Arksey and O'Malley's [19] methodological framework for scoping studies, which was further developed by Levac et al. [20]. The framework has six stages: 1) identifying the research question, 2) identifying relevant studies, 3)

study selection, 4) charting the data, 5) collating, summarising and reporting the results and 6) consultation. The Preferred Reporting Items for Systematic Reviews and Meta-Analysis Extension for Scoping Review (PRISMA-ScR) guidelines [21] was followed. The review protocol was not published but is available on request.

### Stage 1: Identifying the research question

The scoping review set out to answer the following questions:

1. What is the impact of stroke on young adults?

2. What are the focus and the expected outcomes of stroke rehabilitation in young adults?

3. What are young adults' experiences of stroke rehabilitation care in acute and health and social care settings?

Specific operational definitions and search terms were established in relation to the research questions, which included the rehabilitation settings and the rehabilitation scope. These operational definitions and search terms are provided in S1 Table.

### Stage 2: Identifying relevant studies

A comprehensive search strategy was developed to gather the potentially relevant literature published in English over the period 2000–2022. The following databases were searched: Cumulative Index to Nursing and Allied Health Literature (CINAHL), MEDLINE, Excerpta Medica database (EMBASE), PsycINFO, Applied Social Sciences Index and Abstracts (ASSIA), Allied and Complementary Medicine Database (AMED), Physiotherapy Evidence Database (PEDro), Cochrane Library and Web of Science. Grey literature was sought through Google Scholar to identify material from the websites of networks, organisations and charities related to stroke. Hand searches were undertaken on the reference lists of the included articles and grey literature to seek any further relevant items. The full search strategy can be found in the (S2 Table).

### Stage 3: Study selection

The retrieved articles were considered for inclusion if they were relevant to the aims of the scoping review. All types of publications were considered for inclusion i.e. journal articles, guidelines, books and editorials ensuring that the broadest range of literature was accessed. This review followed the approach of searching the wide spectrum of available literature. This scoping review approach focusses on identifying the broadest range of literature available on a topic regardless of the methodological quality of the articles [19]. Articles that did not report on the experiences of young adults (18–45 years) in stroke rehabilitation or did not include stroke patients within this age group in their rehabilitation programme were excluded. Only publications written in English language were included.

All references derived from the search strategy were imported into the Rayyan software for systematic reviews [22]. Two groups of reviewers (AFP and CCl; HC and LA) screened the titles and abstracts against the inclusion and exclusion criteria (Table 1) with reasons for exclusions noted. Consensus between reviewers was reached through discussion.

### Stage 4: Charting the data

Data extraction was performed using a 'descriptive-analytical' method for extracting and summarising the information [19]. General information of each included article was collected to

**Table 1. Inclusion and exclusion criteria.**

| Characteristic | Inclusion criteria | Exclusion criteria |
| --- | --- | --- |
| Participants | Reported the experiences of people diagnosed with stroke aged between 18–45 years | Only Reported the experiences of people diagnosed with stroke aged below 18 and above 45 years or their carers |
| Settings | Reported rehabilitation experiences in acute and social care settings | |
| Publication date | Published between January 2000 and April 2022 | Published before January 2000 and after April 2022 |
| Language | Published in English | Published in languages other than English |
| Type of publication | All type of publications, including journal articles, guidelines, books and editorials | Conference abstracts |

provide context, as well as participants' characteristics, such as age, stroke type and time since stroke onset. Only the subset of data related to young stroke survivors (18–45 years) was extracted in articles where the range of participants' ages extended beyond this.

Articles were classified into two main groups to facilitate data organisation: (1) young adults' general experiences of stroke and rehabilitation care (articles that did not explicitly describe an intervention or rehabilitation programme); and (2) young adults' experiences of particular stroke rehabilitation programmes. For the latter group, the data extraction framework was informed by the logic of realist synthesis using Pawson and Tilley's context-mechanism-outcome (CMO) heuristic tools [23]. Information was collected regarding (1) the programme name, (2) the contexts (C), (3) the underlying mechanisms (M), and (4) the outcomes (O) of rehabilitation for the young stroke survivors [24,25]. Context referred to circumstances related to a stroke rehabilitation programme that had an impact on outcome [26]. Mechanism referred to resources offered by a stroke rehabilitation programme and how stroke survivors responded to these resources [27]. Outcome in this review included both short and long-term outcomes.

As this review aimed to map the existing literature on the lived rehabilitation care experiences of young adults post stroke informed by realist thinking, it did not seek to identify realist programme theories to provide causal explanation of how a programme works or fails in a given context [28]. Instead, the CMO heuristic tools were employed to collate and present the results.

Data extraction was undertaken by two independent reviewers (AFP, CCl) and the results were compared to assess for consistency and resolution of discrepancies by discussion within the core team (AFP, CCl, LA, HRC, CC).

## Stage 5: Collating, summarising and reporting the results

The PRISMA Extension for Scoping Reviews (PRISMA-ScR) checklist was followed in preparing the review report (S3 Table) [21]. A PRISMA diagram [29] was used to summarise the selection process. The review findings are reported and described under three sections that addressed the review questions: (1) the impact of stroke on young adults; (2) the focus and the expected outcomes of stroke rehabilitation for young adults; and (3) young adults' experiences of stroke rehabilitation.

## Stage 6: Consultation

The scoping review was part of a larger research project aiming to explore the range of young adults' experiences following stroke through their own digital accounts and the published

literature, in relation to the formal and informal services available to support them (YARNS project) [30]. The Nursing Studies Ethics Research Panel of the University of Edinburgh approved the study on March 3rd, 2020 (Ref: Staff 173). A patient and public involvement (PPI) group of seven people with personal experience of young stroke advised on all aspects of the YARNS project and acted as 'critical friends'. Stakeholder consultation was undertaken through regular online meetings with the YARNS project partners, research team and the PPI group to provide critical comment related to their experiences on the findings of the preliminary analysis of the scoping review.

## Results

### Search outcome

The study selection process is summarised in Fig 1. From a total of 9320 identified articles, 404 were considered potentially relevant following title and abstract screening. A further 319 articles were excluded after full-text screening, leaving a final result of 85 articles included in this review.

The general characteristics of the included literature are summarised in Table 2.

### Characteristics of included literature

From Table 2 it can be noted that 301 young adults with stroke out of the 1626 total adult stroke sample were identified in the included literature, and 47 of the 85 articles had only one or two young stroke adults in their sample.

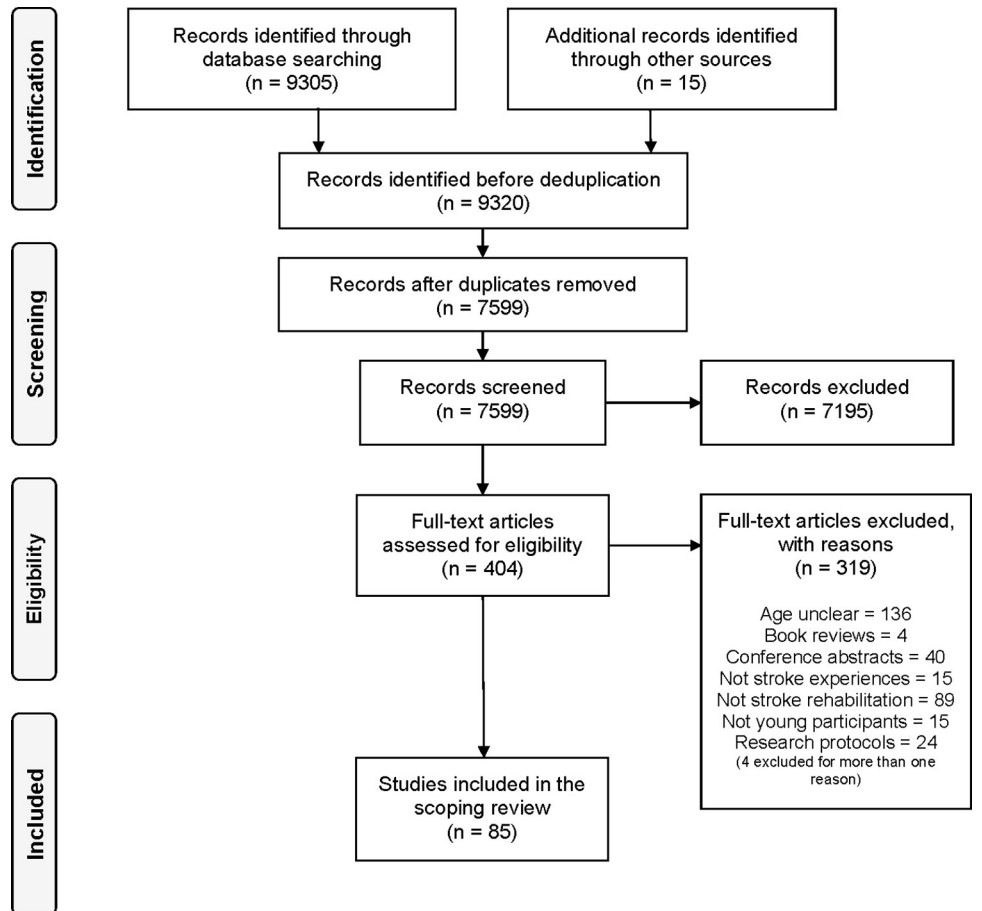

**Fig 1. PRISMA flow diagram for study selection [29].**

**Table 2. Summary of included studies.**

| First Author | Year | Country of Origin | Study Design | Study Aim | Rehabilitation Setting | Stroke Sample All ages* (n) | Stroke Sample Young Adults 18–45** (n) | Young Adults Age Range at time of stroke*** |
|---|---|---|---|---|---|---|---|---|
| Tolat [31] | 2000 | USA | Qualitative | To describe experience and therapeutic drug management during inpatient rehabilitation with three patients with cocaine-associated stroke. | Inpatient rehabilitation | 3 | 1 | 44 |
| Kersten [14] | 2002 | UK | Cross-sectional | To examine the unmet needs of young people with stroke, living in community housing in the UK. | Community | 315 | 51 | 18–45 |
| Immenschuh [32] | 2003 | Germany | Qualitative | To capture the experience of having had a stroke in people under fifty-five during the first year after their stroke. | Inpatient rehabilitation, community | 11 | 9 | 21–45 |
| Kelly [33] | 2003 | Australia | Cross-sectional | To evaluate the cardiorespiratory fitness of subacute stroke patients and to determine whether reduced fitness is associated with gait performance. | Inpatient rehabilitation | 17 | 4 | 24–45 |
| Röding [34] | 2003 | Sweden | Qualitative | To get knowledge of the younger stroke patient's viewpoint and to describe how young stroke patients experience the rehabilitation process. | Community | 5 | 4 | 36–45 |
| Seibert [35] | 2003 | USA | Case study | To investigate strokes' effects on awareness and cognition from the patients' perspective. | Inpatient rehabilitation, community | 4 | 1 | 37 |
| Logan [36] | 2004 | UK | Qualitative | To explore attitudes and barriers to the use of transport with the aim of informing rehabilitation. | Community | 24 | 1 | 43 |
| Murray [37] | 2004 | UK | Qualitative | To investigate the meaning and experience of being a stroke survivor. | Community | 10 | 7 | 18–45 |
| Higgins [38] | 2005 | UK | Qualitative | To investigate the delivery of an arts-based intervention to stroke patients and to seek users' and professionals' views of any perceived benefits. | Inpatient rehabilitation | 21 | 3 | 32–42 |
| Dale Stone [39] | 2005 | UK, USA, Canada | Qualitative | To discover the meaning(s) that survivors attach to their stroke experiences and understand this within the context of their lives and the extent to which there are commonalities and differences in terms of experiences and issues of concern. | Inpatient rehabilitation, community | 22 | 14 | 19–44 |
| Arnaert [40] | 2006 | Canada | Qualitative | To explore the perceptions of hope from patients during the acute care phase of their stroke. | Inpatient rehabilitation | 8 | 2 | 19–37 |
| Holland [41] | 2006 | UK | Qualitative | To describe three individuals who live full and satisfying lives despite aphasia following stroke. | Outpatient rehabilitation | 3 | 1 | 29 |
| Townend [42] | 2006 | UK | Mixed-methods | To investigate fear of recurrent stroke and beliefs about its causes and prevention. | Inpatient rehabilitation, community | 89 | 1 | 42 |
| Anonymous [43] | 2007 | UK | N/A[†] | N/A. | Community | 1 | 1 | 37 |
| Allison [44] | 2008 | UK | Qualitative | To explore the experiences of patients and carers of receiving secondary prevention advice and use these to inform the development of an educational resource. | Primary care | 25 | 2 | 37, 44 |

*(Continued)*

**Table 2.** (*Continued*)

| First Author | Year | Country of Origin | Study Design | Study Aim | Rehabilitation Setting | Stroke Sample All ages* (n) | Stroke Sample Young Adults 18–45** (n) | Young Adults Age Range at time of stroke*** |
|---|---|---|---|---|---|---|---|---|
| Dickson [45] | 2008 | UK | Qualitative | To investigate the beliefs and experiences of people with dysarthria as a result of stroke in relation to their speech disorder, and to explore the perceived physical, personal and psychosocial impacts of living with dysarthria. | Community | 25 | 1 | 32 |
| Jones [46] | 2008 | UK | Qualitative | To learn more about individuals' beliefs and personal strategies used to support the period of recovery after stroke. | Community | 10 | 1 | 29 |
| Ownsworth [47] | 2008 | Australia | Case study | To investigate a participant's perspective of experiences in therapy throughout an awareness rehabilitation intervention. | Outpatient rehabilitation | 1 | 1 | 28 |
| White [48] | 2008 | Australia | Qualitative | To explore the long-term experience of mood changes in community-dwelling stroke survivors at 1, 3, and 5 years after stroke. | Community | 12 | 1 | 40 |
| Catalano [49] | 2009 | N/A‡ | N/A | To explore poststroke experiences, including their primary concerns about their condition, any changes in their health, and their support arrangements in the community. | Community | 6 | 6 | 18–45 |
| Thompson [50] | 2009 | UK | Qualitative | To explore the impact of stroke consequences on spousal relationships from the perspective of the person with stroke. | Community | 18 | 2 | 31, 45 |
| Norris [51] | 2010 | Indonesia | Qualitative | To explore the understanding and perception of stroke in rural Central Aceh; and to identify the mediating factors in that understanding. | Community | 11 | 3 | 32–44 |
| Salisbury [52] | 2010 | UK | Qualitative | To explore in depth the participants lived experiences of a specific phenomenon, namely the healthcare system and services received after stroke. | Inpatient and outpatient rehabilitation | 13 | 2 | 40, 41 |
| Samuel [53] | 2010 | UK | N/A† | N/A. | Inpatient rehabilitation | 1 | 1 | 39 |
| Townend [54] | 2010 | UK | Mixed-methods | To study the association of acceptance of disability with depression following stroke and its ability to predict depression at follow-up. | Inpatient rehabilitation, community | 89 (first month), 81 (ninth month) | 1 | 42 |
| Beesley [55] | 2011 | Australia | Qualitative | To explore the experience of community dwelling stroke survivors' participation in an arts health group programme and possible health benefits to quality of life and wellbeing. | Community | 11 | 2 | 39, 41 |
| Northcott [56] | 2011 | UK | Qualitative | To explore why people lose contact with their friends, and whether there are any protective factors. | Community | 29 | 1 | 18 |

(*Continued*)

**Table 2.** (Continued)

| First Author | Year | Country of Origin | Study Design | Study Aim | Rehabilitation Setting | Stroke Sample All ages* (n) | Stroke Sample Young Adults 18–45** (n) | Young Adults Age Range at time of stroke*** |
|---|---|---|---|---|---|---|---|---|
| Pearl [57] | 2011 | UK | Qualitative | To identify the effects of volunteering activity on people with aphasia and the requirements for the effective involvement in such activity for people with aphasia. | Community | 15 | 3 | 18–45 |
| Prajapati [58] | 2011 | Canada | Case series | To quantify the total time of walking and individual walking bout duration during the course of an inpatient day, compare temporal swing phase symmetry between a commonly used spatiotemporal gait assessment and walking performed throughout the day, investigate the link between characteristics of walking activity and stroke severity. | Inpatient rehabilitation | 16 | 3 | 31–45 |
| Wu [59] | 2011 | USA | Case study | To evaluate whether combined physical and mental practice would increase functional performance and self-perception of performance in a patient with hemiparesis and concomitant ideomotor apraxia after stroke. | Outpatient rehabilitation | 1 | 1 | 44 |
| Kouwenhoven [60] | 2012 | Norway | Qualitative | To describe the lived experience of stroke survivors suffering from depressive symptoms in the acute phase. | Inpatient rehabilitation | 9 | 2 | 30, 45 |
| McCain [61] | 2012 | USA | Case series | To describe the muscle activation patterns and gait characteristics of three persons after stroke who wore an Ankle-Foot Orthosis designed to facilitate typical gait mechanics. | Inpatient and outpatient rehabilitation | 3 | 1 | 44 |
| Middleton [62] | 2012 | N/A‡ | Case study | To describe the case of a young woman with subarachnoid hemorrhage and Terson syndrome through her acute hospital admission, rehabilitation treatment, ophthalmologic management, and outcome. | Inpatient and outpatient rehabilitation | 1 | 1 | 35 |
| Morris [63] | 2012 | UK | Qualitative | To examine stroke patients', carers' and volunteer supporters' experiences of peer support groups during hospital rehabilitation. | Inpatient rehabilitation | 12 | 2 | 43, 44 |
| Norris [64] | 2012 | Indonesia | Qualitative | To explore the subjective experience of stroke in central Aceh. | Community | 11 | 3 | 32–44 |
| Wilkie [65] | 2012 | UK | Qualitative | To explore the impact of Functional Electrical Simulation as applied in the management of dropped foot on patients with chronic stroke and their carers. | Outpatient rehabilitation | 13 | 2 | 40, 41 |
| Gustafsson [66] | 2013 | Australia | Qualitative | To enhance understanding of the transition experience for clients with stroke and their carers during discharge and the first month at home. | Inpatient rehabilitation, community | 5 | 1 | 36 |
| Norris [67] | 2013 | UK | Qualitative | To explore the experience of participation in the action for rehabilitation in Neurological Injury (ARNI) programme. | Community | 22 | 4 | 19–43 |

(Continued)

**Table 2.** (Continued)

| First Author | Year | Country of Origin | Study Design | Study Aim | Rehabilitation Setting | Stroke Sample All ages* (n) | Stroke Sample Young Adults 18–45** (n) | Young Adults Age Range at time of stroke*** |
|---|---|---|---|---|---|---|---|---|
| Saito [68] | 2013 | Japan | Qualitative | To illustrate the importance of the cooperation between medical institutions and work support agencies and discussed reasons why medical institutions have difficulties in supporting persons who have experienced a stroke in their return to work. | Outpatient rehabilitation, community | 2 | 1 | 29 |
| Casey [69] | 2014 | Canada | Case study | To measure the effect of 12-week water-based exercise programme on cardiovascular fitness, balance, motor control, and mobility. | Community | 1 | 1 | 18 |
| Connell [70] | 2014 | UK | Qualitative | To gain an insight into how stroke survivors experience somatosensory impairment after stroke. | Outpatient rehabilitation | 5 | 1 | 44 |
| Gustafsson [71] | 2014 | Australia | Qualitative | To investigate the experiences and expectations of people with stroke, during their transition from hospital to home, after participating in a novel inpatient outreach program, entitled STRENGTH. | Inpatient rehabilitation, community | 7 | 1 | 36 |
| Kirkevold [72] | 2014 | Norway | Qualitative | To evaluate the content, structure and process of a dialogue-based psychosocial nursing intervention in primary care and its usefulness from the perspective of stroke survivors. | Inpatient rehabilitation, community | 25 | 2 | 33, 43 |
| Kuluski [73] | 2014 | UK | Qualitative | To understand the experience of stroke as a disabling life situation among young people and the strategies that they use to recover and cope. | Community | 17 | 12 | 21–45 |
| Pallesen [74] | 2014 | Denmark | Qualitative | To identify, from a long-term perspective, stroke survivors' self-identity, their views of any associated disabilities and how they manage their lives after stroke. | Community | 15 | 2 | 40–42 |
| Sadler [75] | 2014 | UK | Qualitative | To investigate how younger stroke survivors' experiences of care are shaped by the field of stroke, and how, in navigating stroke care, individuals seek to draw on different forms of capital in adjusting to life after stroke. | Inpatient rehabilitation, community | 27 | 10 | 24–44 |
| Armstrong [76] | 2015 | Australia | Qualitative | To explore consequences of acquired communication disorders for Aboriginal Australians after stroke, including their experiences of services received. | Community | 13 | 6 | 20–43 |

(*Continued*)

**Table 2.** (*Continued*)

| First Author | Year | Country of Origin | Study Design | Study Aim | Rehabilitation Setting | Stroke Sample All ages* (n) | Stroke Sample Young Adults 18–45** (n) | Young Adults Age Range at time of stroke*** |
|---|---|---|---|---|---|---|---|---|
| Arntzen [77] | 2015 | Norway | Qualitative | To develop a deeper understanding of how the dynamic phenomenon body, participation in everyday life and sense of self interrelates and changes through stroke survivors' movement in and between the two contexts and what this phenomenon means for stroke survivors' process of change and well-being in the early rehabilitation trajectory. | Inpatient rehabilitation, community | 9 | 3 | 31–42 |
| Ferrarin [78] | 2015 | Italy | Observational | To assess the impact of gait analysis on clinical decision-making in adult chronic poststroke patients. | Inpatient rehabilitation | 49 | 17 | 24–45 |
| Martinsen [79] | 2015 | Norway | Qualitative | To explore young and midlife stroke survivors' experiences with health services and to identify long-term follow-up needs. | Community | 16 | 7 | 19–43 |
| Morris [80] | 2015 | UK | Qualitative | To explore and compare the views of stroke survivors, carers and physiotherapists about physical activity after rehabilitation to examine the contextual factors that are perceived to influence survivors' physical activity participation. | Community | 38 | 1 | 42 |
| White [81] | 2015 | Australia | Qualitative | To qualitatively explore stroke survivors' experience of implementation of exposure to an enriched environment u within a typical stroke rehab setting, in order to identify facilitators and barriers to participation. | Inpatient rehabilitation | 10 | 1 | 41 |
| Wolfenden [82] | 2015 | Australia | Qualitative | To explore the experiences of young higher functioning stroke survivors in re-establishing identity and returning to work | Community | 5 | 5 | 28–44 |
| Gorst [83] | 2016 | UK | Qualitative | To explore the nature and impact of foot and ankle impairments on mobility and balance in community-dwelling, chronic stroke survivors. | Community | 13 | 2 | 32, 41 |
| Leahy [16] | 2016 | Ireland | Qualitative | To explore the experience of stroke among young women. | Community | 12 | 1 | 22 |
| Moorley [84] | 2016 | UK | Qualitative | To identify the coping mechanisms that African Caribbean women used post stroke and the implications of these coping mechanisms for stroke recovery and lifestyle modification efforts. | Community | 7 | 2 | 29, 40 |
| Rosewilliam [85] | 2016 | UK | Qualitative | To explore whether goal-setting for rehab with acute stroke survivors is patient-centred and identify factors which influence the adoption of patient-centredness in goal-setting practice. | Inpatient rehabilitation | 7 | 1 | 42 |
| Lou [86] | 2017 | Denmark | Qualitative | To investigate how mild stroke patients' and their partners' experience and manage everyday life in a context of early supported discharge. | Inpatient rehabilitation, community | 22 | 1 | 41 |

(*Continued*)

**Table 2.** (Continued)

| First Author | Year | Country of Origin | Study Design | Study Aim | Rehabilitation Setting | Stroke Sample All ages* (n) | Stroke Sample Young Adults 18–45** (n) | Young Adults Age Range at time of stroke*** |
|---|---|---|---|---|---|---|---|---|
| Mavaddat [87] | 2017 | UK | Qualitative | To assess acceptability of training in positivity with Positive Mental Training (PosMT) for prevention and management of post-stroke psychological problems and to help with coping with rehabilitation. | Community | 10 | 1 | 34 |
| Sieber [88] | 2017 | USA | Qualitative | To explore the author's personal experiences and how the author understood them at the time and reflect on artifacts to find meanings and themes associated with her experience. | Community | 1 | 1 | 24 |
| Walder [89] | 2017 | Australia | Qualitative | To understand the emerging picture of occupational disruption and identity reconstruction after stroke. | Community | 6 | 2 | 33, 45 |
| Palstam [90] | 2018 | Sweden | Qualitative | To explore how persons experienced return to work and their work situation 7 to 8 years after a stroke. | Community | 13 | 3 | 31–41 |
| Shipley [91] | 2018 | Australia | Qualitative | To examine the personal and social experiences of younger adults after stroke. | Community | 19 | 15 | 19–44 |
| Smith [92] | 2018 | UK | Qualitative | To explore the experiences of both stroke survivors and caregivers and secondly identify their ongoing needs. | Community | 3 | 1 | 44 |
| Valkenborghs [93] | 2018 | Australia | Case series | To describe the exploratory implementation of a combined aerobic exercise and task-specific training intervention to improve upper limb motor function in one person in subacute stroke recovery and one person in chronic stroke recovery. | Inpatient rehabilitation | 2 | 1 | 35 |
| Abrahamson [94] | 2019 | UK | Qualitative | To explore the contribution of the sixth-month review to overall recovery for patients and carers. | Community | 46 | 1 | 37 |
| Dwyer [18] | 2019 | Ireland | Qualitative | To explore the lived experience of young adults with brain injury residing in aged care facilities. | Nursing home | 6 | 1 | 43 |
| Hodson [95] | 2019 | Australia | Qualitative | To explore the experience of people with mild stroke in the first 6 months after hospital discharge. | Community | 5 | 2 | 40, 45 |
| Jarvis [96] | 2019 | UK | Cross-sectional | To investigate how a stroke in young adults affects walking performance (e.g., walking speed and metabolic cost) compared with healthy age-matched controls. | Inpatient rehabilitation | 46 | 6 | 20–45 |
| Pallesen [97] | 2019 | Norway, Denmark | Qualitative | To explore stroke patients' experiences of rehabilitation pathways in Norway and Denmark. | Community | 11 | 2 | 40, 45 |
| Theadom [98] | 2019 | New Zealand | Qualitative | To explore people's experiences over the first three years post-stroke and identify what helped or hindered recovery. | Community | 55 | 4 | 42–43 |
| Törnbom [99] | 2019 | Sweden | Qualitative | To explore participants' experiences in everyday life after stroke and potential aspects of participation through the photovoice method. | Community | 11 | 1 | 31 |

*(Continued)*

**Table 2.** (Continued)

| First Author | Year | Country of Origin | Study Design | Study Aim | Rehabilitation Setting | Stroke Sample All ages* (n) | Stroke Sample Young Adults 18–45** (n) | Young Adults Age Range at time of stroke*** |
|---|---|---|---|---|---|---|---|---|
| Törnbom [100] | 2019 | Sweden | Qualitative | To enhance the understanding of long-term participation in working-aged people 7–8 years after stroke. | Community | 11 | 7 | 25–44 |
| White [101] | 2019 | USA | Mixed-methods | To describe control of risk factors after stroke from the perspectives of the stroke survivor, the family, and healthcare professionals. | Community | 18 (qualitative) | 1 | 43 |
| Wilson [102] | 2019 | UK | N/A† | N/A. | Inpatient rehabilitation | 1 | 1 | 42 |
| Bailey [103] | 2020 | USA | Qualitative | To examine outcomes expectations, self-efficacy, self-regulation, and social support for daily physical activity among study participants. | Community | 15 | 2 | 40, 41 |
| Pereira [104] | 2020 | Portugal | Qualitative | To understand how people with stroke and carers adapt over time, and how health professionals support transition to home. | Outpatient rehabilitation | 8 | 1 | 43 |
| Shipley [8] | 2020 | Australia | Qualitative | To examine the unmet needs of younger stroke survivors in inpatient and outpatient healthcare settings and identify opportunities for improved service delivery. | Community | 19 | 15 | 19–44 |
| Masterson-Algar [105] | 2020 | UK | Qualitative | To codesign and construct a peer-led coaching intervention to improve leisure and social participation after stroke. | Community | 79 | 2 | 40, 41 |
| Millard [106] | 2020 | Australia | Qualitative | To understand the experiences of people use SaeboFlex® in an outpatient setting | Outpatient rehabilitation | 5 | 1 | 39 |
| Purcell [107] | 2020 | Australia | Qualitative | To explore stroke survivors experiences of engagement in occupations during stroke rehabilitation. | Inpatient | 8 | 1 | 45 |
| Vaughan-Graham [108] | 2020 | Canada | Qualitative | To explore in-depth, end-users perspectives, persons with stroke and physiotherapists, following a single-use session with a H2 exoskeleton. | Outpatient rehabilitation | 5 | 1 | 38 |
| Walder [109] | 2020 | Australia | Qualitative | To explore how stroke survivors perceive their relationship with their health care team as they adjust to life following stroke. | Outpatient rehabilitation | 6 | 2 | 34, 45 |
| Withiel [110] | 2020 | Australia | Qualitative | To obtain in-depth feedback from participants about their qualitative experience in different memory interventions to better understand factors that may impact outcomes. | Community | 20 | 1 | 32 |
| Panda [111] | 2021 | Australia | Qualitative | To explore the impact of meditation for people with chronic aphasia. | Community | 5 | 1 | 44 |

*Only adults with stroke. Other study participants such as carers and staff were not included in the total number.

** The number of young adults (18–45) included in the All age adult sample.

***If not indicated, we assumed that the ages mentioned in the article were at the time of stroke.

†Not a research study, either magazine/online article or a book chapter.

‡Location not reported.

The summary characteristics of included literature are presented in Table 3. Most studies originated from the United Kingdom (n = 31, 36.5%) and Australia (n = 19, 22.4%). Of the 85 articles, 39 (45.9%) were published between 2015 and 2022, suggesting a growing interest in the topic area. Qualitative research studies made up over three quarters of the articles (n = 66, 77.7%). Stroke rehabilitation in the studies was delivered mainly in community settings (n = 45, 52.9%).

## The impact of stroke on young adults

The impact of stroke on young adults was reported in 39 articles. Both negative and positive consequences were identified. These results are presented in two groups: (1) studies reporting negative consequences and (2) studies reporting positive consequences.

**Studies reporting negative consequences.** The negative consequences of stroke on young adults are presented under the following categories: physical, financial, social and psychological impacts.

*Physical impacts.* The most commonly reported physical impact of stroke was related to changes in body and brain function, such as movement limitations [9,32,74,83], communication problems [35,41,45,76,104] and memory deficits [35]. The influence of physical problems on performance and function seemed to be construed by young adults as a loss of self-identity as they expressed that they were not the same person following stroke [9,16,32,37,77,91]. Several young adults also spoke of their worry that physical disability would affect how other people perceived their body appearance [39,83]. When considering overall differences, participants explained how their energy levels had reduced after having a stroke. They said they "feel tired all the time" or one said, "I feel like I could just lie down all day" [105].

*Financial impacts.* Young adults who had no financial security reported financial consequences following stroke. They suffered from loss of income due to being unable to return to work [9,35,41,64,75,82,91,95]. Several young adults said that they were not able to access income support due to their invisible disabilities and that they were deemed able to work [82]. As a result, many of them relied on personal savings and financial support from their family [82]. Importantly, young adults attributed their low attendance at rehabilitation programmes to their financial hardships [82].

*Social impacts.* The social consequences of stroke in young adults were found in their relationships with their spouses or families, within their social circles and with the wider society. Stroke was deemed as altering marital dynamics resulting from a loss of sexual functioning [37,50] and role change of the stroke survivor from spouse to care recipient [9,50]. Marital or partner relationship disruptions were frequently reported in the literature [37,49,64,91,95]. Some examples were a man being left by his wife after his stroke [91] and a breakup between a young woman with a baby and her fiancé following her stroke [49]. An ethnographic study also found that husbands were more likely to leave if their wives had a stroke [64]. These findings further highlighted stroke's profound impact on marriage and relationships.

The reported impact of stroke on family dynamics was related to physical disability, emotional changes and communication issues following stroke. One young man explained that his stroke limited his ability to interact with his little children, such as playing or carrying them [9]. Several young adults mentioned that they became easily irritated post-stroke and often exhibited anger and temper towards their children and family members [37,91]. A young woman with dysarthria was frustrated when her children silently watched her when she was speaking [45]. Another young adult said she had to send her children away when they were causing her stress and this impacted on her parenting style [105].

**Table 3. Summary characteristics of included literature.**

| Characteristic | Number (n) | Percentage (%) |
|---|---|---|
| Country of origin | | |
| Australia | 19 | 22.4 |
| Canada | 4 | 4.7 |
| Denmark | 2 | 2.4 |
| Germany | 1 | 1.2 |
| Indonesia | 2 | 2.4 |
| Ireland | 2 | 2.4 |
| Italy | 1 | 1.2 |
| Japan | 1 | 1.2 |
| New Zealand | 1 | 1.2 |
| Norway | 4 | 4.7 |
| Norway and Denmark | 1 | 1.2 |
| Portugal | 1 | 1.2 |
| Sweden | 4 | 4.7 |
| UK | 31 | 36.5 |
| USA | 8 | 9.4 |
| Multiple countries (>3 countries) | 1 | 1.2 |
| N/A (not reported) | 2 | 2.4 |
| Publication year | | |
| 2000–2004 | 8 | 9.4 |
| 2005–2009 | 13 | 15.3 |
| 2010–2014 | 25 | 29.4 |
| 2015–2022 | 39 | 45.9 |
| Publication type | | |
| Book | 1 | 1.2 |
| Original study | 79 | 92.9 |
| Magazine article | 2 | 2.4 |
| Thesis | 2 | 2.4 |
| Website article | 1 | 1.2 |
| Study design | | |
| Case series | 3 | 3.5 |
| Case study | 5 | 5.9 |
| Cross-sectional | 3 | 3.5 |
| Mixed-methods | 3 | 3.5 |
| Observational | 1 | 1.2 |
| Qualitative | 66 | 77.7 |
| N/A (not a research study) | 4 | 4.7 |
| Rehabilitation setting | | |
| Community | 44 | 51.8 |
| Community, outpatient rehabilitation | 1 | 1.2 |
| Inpatient | 3 | 3.5 |
| Inpatient rehabilitation | 12 | 14.1 |
| Inpatient rehabilitation, outpatient rehabilitation | 1 | 1.2 |
| Inpatient, community | 6 | 7.1 |
| Inpatient, outpatient rehabilitation | 2 | 2.4 |
| Inpatient rehabilitation, community | 5 | 5.9 |
| Nursing home | 1 | 1.2 |

*(Continued)*

**Table 3.** (Continued)

| Characteristic | Number (n) | Percentage (%) |
|---|---|---|
| Outpatient rehabilitation | 9 | 10.6 |
| Primary Care | 1 | 1.2 |

Young adults tended to refrain from socialising activities after stroke. They reported that they got exhausted quickly [9,74,100] and they would avoid crowds and noisy places [37,89]. However, having less social interaction could lead to feeling isolated [74,91]. In one case, a man mentioned that he had invited some unknown callers to come inside his house in his desperation to talk to someone after being discharged from the hospital [91]:

> *When I had the stroke and came home. . . nobody came. . . I was that desperate to talk to people, and I know it sounds awful this, but I got a knock on the door by [some unkown callers] and I invited them in for a cup of tea and a chat because they were the only people that came.* [91, p6]

Social interaction could be challenging for young adults following stroke. Invisible disabilities, particularly cognitive disabilities, were perceived as being dismissed by other people [37,39]. Some young people also reported that people mistakenly assumed that they were drunk [37]. They also reported their experiences of being socially stigmatised, as it was assumed that their stroke was caused by cigarette smoking or drug use [91].

Young adults following stroke felt stigmatised by a condition that was expected only in older adults. They reported disbelief regarding their stroke diagnosis, claiming that stroke was not supposed to affect young and fit people with healthy lifestyles [16,32,39]. Young adults seemed not to associate the symptoms they experienced as a sign of stroke, nor were they aware that they had had a stroke [32,52,89]. Some examples included dismissing slurred speech as a sign of tiredness, rather than stroke [32], and rejecting a friends' advice to go to hospital and going home instead [89].

Young adults who sought help at the hospital were often asked to go home by the healthcare staff as their stroke signs were misinterpreted, resulting in delayed treatment. Some examples included severe headache being diagnosed as an inner ear infection and vomiting as being caused by drug use [82]. A young adult reported that she given another appointment in a week despite the computerised tomography (CT) scan confirming her stroke [32]:

> *And he [the radiologist] held up some x-rays and marked something and said that was a blood, what do you call it? A blood clot. And "You had a stroke. We'll make an appointment for next week" and of course I went "Yes, sure" I was totally out of it, hadn't grasped what he just had told me and then I took the CT- pictures. . .and "OK, see you next week then" and I went out and I was there with a friend and told her everything and she went "Hey, XXX! Why on earth are you still here? I thought you'd had a stroke, don't you have to get taken in to hospital or something?" "No idea."* [32, p155]

*Psychological impacts.* Some young adults spoke of the psychological impacts of stroke, their fear of recurrent stroke [32,42,91] and death [18,43]. Many young people expressed their grief over their stroke diagnosis with common feelings of denial [9,32,39,51,91,95], bargaining [77], and depression [39,43,49,91]. Several young adults described their grief as a sudden loss of their active self or the death of their old self [9]. Bargaining in young adults was related to an expression of hope that their physical limitations after stroke could be 'repaired' [77]:

*eh. . . right now, this hand is not supposed to function. It will require help from the left hand. . .so there were many such things that–That you in some way became used to. . . strangely enough. And it probably led to that I felt that–that things worked well.* [77, p310]

Depression appeared to be a frequent psychological issue in the literature among the young adults following stroke, with anti-depressants prescribed to deal with their depression [43,47,77] Young adults also conveyed their frustration associated with their whole stroke journey. The frustration was apparent when they received their first stroke diagnosis, returning home, performing personal care activities or daily routines, and maintaining their relationships [8,9,35,37,48,66,88,91]. A few young adults reported experiencing fatigue when performing light jobs or daily tasks at work [35,50,100]. They also expressed that their reliance on significant others, or other people, post stroke for everyday activities led to feeling vulnerable [9,32,37,41,46,64,76,82,102]

**Studies reporting positive consequences.** Positive impacts of stroke were reported in 11 of the 85 articles. Young adults said that they appreciated life more following stroke as they were still alive [9,32,100]. Stroke was also considered as shifting their life priorities to focus more on their well-being, some examples being through relaxing activities such as handicrafts and cuddling with pets [99], and learning to practice their spiritual gratitude [32]. Having more time with their spouse and children was seen as a positive impact of stroke [9,49,80], as well as being more engaged with their community [49]. Moreover, young adults shared how they developed personal strength throughout their stroke journey [9,32,41,76,84,88,91], for instance, through humour [91] and by keep pushing and trying to overcome obstacles that they encountered [84,88].

## The focus and the expected outcomes of stroke rehabilitation for young adults

In the 85 included articles, only 29 of them specifically evaluated existing rehabilitation programmes [31,33,38,40,44,47,55,57–59,61–63,65,67–72,78,81,86,87,89,92,93,101,103]. The findings of the young adults' experiences are presented in the context-mechanism-outcome (CMO) configuration and summarised in Table 4.

In general, stroke rehabilitation programmes focused on three main outcomes, physical, psychological and wellbeing of stroke survivors. These were categorised into four different types of programme, (1) physical, (2) psychological and well-being, (3) neuro-pharmacological and (4) integrated.

Physical related programmes included stroke rehabilitation programmes that focused on regaining strength in movement, sensorimotor function of upper and lower limb, aerobic endurance, balance, and coordination [33,58,61,62,65,67,69,70,78,92,93,103]. The expected outcomes were improved strength, balance, and mobility. They measured indicators such as stroke survivors' velocity, endurance, and peak oxygen uptake. The tools used to measure motor functions were the action research arm test (ARAT) and Wolf motor function test (WMFT). The stroke impact scale (SIS) was employed to measure activities of daily living, mobility, and participation. Although the focus from these types of intervention is physical outcomes, it should be noted that the social aspect of the programmes, such as interactions with peers, serve as a mechanism that brings positive impacts on the psychological state of young adults, for instance, increased self-confidence. Participants also highlighted three limitations of physical-related programmes in stroke rehabilitation, including short duration, late start of the programme, and lower priority on sensory issues.

Psychological related stroke rehabilitation programmes addressed psychological and social problems as well as promoting the well-being of stroke survivors, and included

**Table 4. Studies of evaluations of programmes for stroke rehabilitation of young adults (n = 29).**

| Programme | Stroke Sample All ages* (n) | Stroke Sample Young Adults 18–45** (n) | Context | Mechanism | | Outcome |
|---|---|---|---|---|---|---|
| | | | | Resource | Response | |
| **Physical-related programme** | | | | | | |
| Somatosensory training [70] | 5 | 1 | Attitudes, thoughts, and motivation. | Trains proprioception of upper and lower limbs. | Lack of sensation, early sensory re-education, regretted that sensation was not a priority in rehabilitation programme | Unable to deal with sensory issues. There is still a need for evidence-based and practice-appropriate clinical assessment tools and treatment strategies to be identified. |
| 12-week land and water-based programme ([69] | 1 | 1 | Age, down syndrome, stroke type, training engagement, parent's support, goal attaining scale. | Trains strength and balance, facilitates social interactions with similar aged peers. | Motivation, physical performance without help, better in socialising. | Improved cardiorespiratory fitness, strength, balance, and mobility. Indicators: oxygen uptake (VO2) peak, 8% increase, one-repetition maximum (51%), community balance and mobility scale (54%), comfortable walking speed (42%), six-minute walk test (28%), daily step count (21%). Improved social behaviour |
| Ambulatory monitoring using Accelerometery for Bilateral Lower Extremities (ABLE) [58] | 16 | 3 | Gender, age, type of stroke, days post-stroke, mobility, motor recovery, gait speed, balance, symmetry | Guides therapists to conduct appropriate changes to therapy, provides measures to monitor treatment outcomes, and serves as homework checker to ensure prescribed daily walking are conducted | Higher walking activity | Increased walking activity and walking bout durations. Significant association was found between the number of walking bouts to total walking time and laboratory gait speed and between laboratory gait speed and balance impairment. Increased in gait asymmetry during day-long measurement compared with the standard laboratory-based assessment |
| Combined aerobic exercise and task-specific training [93] | 2 | 1 | Adherence to protocol, suitable for subacute and chronic phase stroke patient | Combines two training, including aerobic exercise of 30 minutes of lower limb cycling and 30 minutes of upper limb task-specific training | The changeover from aerobic exercise to task-specific training interrupted the flow of the session and reduced recovery potential, the programme was too short | Upper limb motor function improved on Action Research Arm Test (ARAT) by 4 points and Wolf Motor Function Test (WMFT) 5 points, aerobic fitness improved 4.66ml/O2/kg/min and 6-minute walking distance 50-meter, increased strength and function in the upper limb, increased participation in daily activities, increased in activities of daily living categories, feeling fitter, increased in education of compensatory movements, and social interaction, provided hope and optimism that they could participate in meaningful activities they had not participated in since their stroke. |

*(Continued)*

**Table 4.** (Continued)

| Programme | Stroke Sample All ages* (n) | Stroke Sample Young Adults 18–45** (n) | Context | Mechanism | | Outcome |
|---|---|---|---|---|---|---|
| | | | | Resource | Response | |
| Ankle-Foot Orthosis (AFO) [61] | 3 | 1 | Weight, height, and comorbidities | Facilitates a long-term motor recovery | N/A | Gait endurance and velocity increased, motor recovery improved, general symmetry improved, muscle activity activate early, amplitude increased |
| Cardiorespiratory exercise [33] | 17 | 4 | Body mass, stroke types and location, comorbidities | Trains cardiorespiratory fitness with gait performance | N/A | Peak oxygen uptake (VO2peak) was 1.150.36L/min, which was only 50% of the VO2peak reported in the literature for a healthy, age-matched group, maximal walking velocity (1.020.28m/s) and endurance (294.1120.2m) were also approximately 50% of an aged-matched healthy group, 6-minute walking endurance was strongly associated with self-selected walking velocity. |
| Gait analysis [78] | 49 | 17 | Walking ability, musculoskeletal condition | Provides spatio-temporal parameters | Gait analysis confidence level was improved | Changes in gait analysis (GA). GA significantly influences therapeutic planning surgical and non-surgical for chronic post-stroke patients with locomotor disability. |
| Rehabilitation programme (no assigned name given) [62] | 1 | 1 | Type of complication in stroke: Terson syndrome | Facilitates patient to effectively implement the necessary physical therapy, occupational therapy, speech therapy skills, and surgery. | N/A | Vision improved, patient successfully return to community and independence with all basic activities daily living |
| Upper limb rehabilitation [92] | 3 | 1 | Timing to start upper-limb rehabilitation | Failed to give early upper-limb intervention as in the community the rehabilitation emphasises more in lower-limb | If the programme started early, the participant felt that they could get more input on arms | Participants of the study felt depressed and loss of independence because of upper-limb impairment |
| Physical activity (PA) [103] | 15 | 2 | Individual factors: age, sex, race, years since stroke, Body Mass Index (BMI), Barthel index, ambulation, physical activity level, expectations, self-efficacy, self-regulation, social support | Encourages stroke survivors to complete activities of daily living, ambulate with or without an assistive device, perform tasks that required physical activity, and repeat practice of challenging tasks (physically active) | Gives a reward feeling of accomplishment, modifies task to compensate the difficulties (a sense of problem solving) | Regained strength, improved performance, strengthened one's self efficacy |
| Functional electrical stimulation (FES) [65] | 13 | 2 | Gait velocity at setup, time since FES setup, home and social circumstances, time since stroke | Produces movement in muscles paralysed due to central nervous system lesions (e.g., lifting the foot during the swing phase of gait). | The quality and stamina of walking is much better and quicker | Improvements in walking, being involved in the family tradition, the positive feelings |

(*Continued*)

**Table 4.** (Continued)

| Programme | Stroke Sample All ages* (n) | Stroke Sample Young Adults 18–45** (n) | Context | Mechanism | | Outcome |
|---|---|---|---|---|---|---|
| | | | | Resource | Response | |
| Action for Rehabilitation in Neurological Injury (ARNI) [67] | 30 | 4 | Peer led and supported, group motivation, terminology (language), environment | Provides one-to-one time to address personal goals | Having opportunities to increase their individual capacity, feeling appreciated, regain independence, seek another activity, increase accessibility because the training was held at a community facility that did not require appointment | Feeling challenged, hard work, returning to activities and roles ceased since their stroke, participating more in community life |
| **Psychological and well-being related programme** | | | | | | |
| Dialogue-based intervention [72] | 25 | 2 | Language problems, group/individual intervention, physical limitations, fatigue, vision or hearing deficiencies, reduced memory, concentration difficulties | Uses work sheets, offers two meeting first meeting occurred as soon as possible after the stroke, usually within 4–8 weeks, and the last occurred approximately 6 months after the stroke (except for the aphasia group, in which the intervention had to be prolonged). | The work sheet was understandable but difficult to read and write, the content was very good, the intervention should last longer, the intervention should be personalised based on the need of participants since the age range was high | Having difficulties to read and write, using the workbook to start thinking, wanting a longer intervention, a mismatch between personal needs and the group topic discussion |
| Art health group programme [55] | 11 | 2 | Aphasia or underlying cognitive impairment | Provides an opportunity for stroke survivors to explore art in a supportive environment | Feeling of accomplishment, provide insight into physical capabilities, increased self-confidence and self-esteem, life-style benefit, change and instil hope, found something new to learn | Improved quality of life and well-being, included increased confidence, self-awareness, and social interaction which lead to improved self-efficacy of participants |
| Storytelling [40] | 8 | 2 | Age, gender, marital status, level of education, type of stroke, stroke severity, presence of sequelae, and number of days after the stroke event | Facilitates participants to share their story in detail about physical symptoms, emotional and social impact of their stroke experience and to discover their vision of hope and its role in life | N/A | Nurtures positive type of hope (active) |
| Positive mental training (PosMT) [87] | 10 | 1 | Level of depression, anxiety, suicidality, affective, and disability | Facilitates relaxation, manage anxiety, regain confidence, coping | Helps participants into a routine, deal with anxiety, sleep deprivation, stressed, panic attack, relaxed, and regain self confidence | Positive physical and psychological benefits, including improved relaxation, better sleep, reduced anxiety, gained positive outlook on the future, increased motivation, confidence, and ability to cope with rehabilitation |

(*Continued*)

**Table 4.** (Continued)

| Programme | Stroke Sample All ages* (n) | Stroke Sample Young Adults 18–45** (n) | Context | Mechanism | | Outcome |
|---|---|---|---|---|---|---|
| | | | | Resource | Response | |
| Self-awareness intervention [47] | 1 | 1 | Previous working experience, persisting awareness deficits, fluctuating emotional state, and motivation for treatment | Provides knowledge of the brain and brain injury, awareness of deficits and their everyday impact, self-evaluation of physical, cognitive and behavioural abilities, gives feedback, emotional support, provides counselling on how to do self-monitoring, identification of goals | Very interesting, gain useful insights and feedback on capability, learning through practical experience, individualising therapy | Participant was offered part-time paid work as a retail assistant (3 days per week, 5 hours per week) and had maintained this position at follow-ups conducted at 3, 6, and 9 months post intervention |
| Hospital-based peer support groups [63] | 8–18 participants (varies across weeks range) | 2 | Laterality, dysphasia (communication problems), mobility, cognition difficulties, with young children, the group size | Provides a media to talk about particular problem | Want to know more about the effect on younger children when their parent has a stroke, feel more positive and encouraging, taught how to listen to others and respond to them | Gained helpful information and advice, built connections, and increased awareness of stroke |
| Arts in health [38] | 21 | 3 | Patients' socio-demographic details, including socio-economic status (SES) using the Registrar General's occupational codes, cognitive status using the abbreviated mental test | Slow pace of the reading sessions, reader/patient relationship, relief from anxiety | Easy to understand, thankful for willing to wait and not being in a rush in telling a story | The sense of being in control and the practical experience of communication, being able to talk freely, confiding things which they felt unable to share with friends and family, an entertaining distraction in a boring and anxiety provoking situation |
| Reconstructing an occupational identity [89] | 6 | 2 | Number of strokes, hospital length of stay, participation in inpatient rehabilitation, outpatient rehabilitation, living situation, working prior to stroke, leisure occupations prior to stroke, ethnicity | Reflecting the impact on their identity through leisure occupations, trying to make sense of symptoms, communication difficulties, discharged from services bringing a confrontation with the reality of the stroke, reframe thinking, re-evaluating priorities, managing emotions | N/A | Feeling destroyed, putting the symptoms down to being tired, hot or unwell, resisting friends' urges to go to hospital, hindering connecting with the reality of the stroke, not realising having the challenges, a sudden sense of isolation at discharge, being grateful that they could return to previous occupation, accepting that the stroke had occurred and looking towards a new future reality, being judged by people when having social interactions |
| Enriched environment [81] | 10 | 1 | age, gender, first ever stroke, length of stay in rehabilitation, discharge destination, mobility restrictions | enhances social interaction, increases activity levels at patient's bedside and the experience of access to activities from a participant's home settings facilitated adaptation to the unfamiliar hospital environment, set daily routines on the ward | Compromised personal preferences towards accessing the communal enriched environment | Feeling constrained and unable to move around the ward at their leisure, feelings of boredom staying at bedside |

*(Continued)*

**Table 4.** (*Continued*)

| Programme | Stroke Sample All ages* (n) | Stroke Sample Young Adults 18–45** (n) | Context | Mechanism | | Outcome |
|---|---|---|---|---|---|---|
| | | | | Resource | Response | |
| Volunteering [57] | 14 | 1 | Level of engagement, according to the following factors: length of time of involvement; regularity of activity; range of activities; number of activities; role within the activities and effects within the activity for the organisations involved, fatigue, other life priorities; time since the stroke; skills possessed prior to having aphasia and those limited by aphasia; concentration, emotions, memory; personality characteristics | Participation in the activity, enabling them to fulfil their self-expectations and live their lives in a personally meaningful way, delivering public presentations, engaging in group discussions, managing conversations, assistance and support given for people with disabilities, activities or services offered | More confident, speak better, give opinion, can relate to someone else | Feeling more confident, enduring personality traits, improvements in communication, family member's support, having empathy with others living with a similar disability, aphasia 'made real' for staff |
| Secondary prevention advice [44] | 25 | 2 | Ability to comprehend information, language used by clinicians to give explanation | Fails to identify the appropriate moment in delivering information, fail to use universal language that can be understood by laypersons | Information given was irrelevant, participant cannot understand the Latin words which were used by doctors | Rejection of information, confusion and misunderstanding |
| Self-management of risk factors [101] | 100 | 1 | Age, gender, race/ethnicity, education, insurance, mean systolic and diastolic blood pressure, medication adherence, know target of blood pressure (BP), ownership of mobile phone, the use of internet to access health information, ownership of home monitors to measure BP or taking their BP at a pharmacy, motivation to learn to take their BP, lack of financial resources | The use of some form of health information technology in supporting risk factor control, such as appointment reminders, instructions from a health professional in the correct procedures for monitoring BP | N/A | Raised awareness of risk, motivated to make changes, searching for information about risk factors on the internet, setting an alarm on their mobile phone as a reminder to take pills, text messages for appointment reminders, home BP monitoring, and using internet sites to track BP over time. |
| Early supported discharge (ESD) [86] | 22 | 1 | Employment status, type of stroke, days in hospital, visits from ESD team, future ESD visits or phone calls planned, participating in community-based rehab | Individual assessment, team visit, evaluation of patients' needs and outline a rehabilitation plan | Home as calm, participant did not feel that the plan covered rehabilitation needs. Participant felt that the services were more suited for retired and older patients. | There was a mismatch between the needs of young adult participants and service offered by ESD |
| **Neuropharmacological related intervention** | | | | | | |

(*Continued*)

**Table 4.** (Continued)

| Programme | Stroke Sample All ages* (n) | Stroke Sample Young Adults 18–45** (n) | Context | Mechanism | | Outcome |
| --- | --- | --- | --- | --- | --- | --- |
| | | | | Resource | Response | |
| Methylphenidate and bromocriptine [31] | 3 | 1 | Drug dose, stroke severity level, other drug uses | Improve neuro function | Participant reported minimal memory and word-finding deficits and mild personality changes | Associated with an excellent functional gain: 50 Functional Independence Measure (FIM) points in 37 days. Patient returned to work. |
| **Integrated programme** | | | | | | |
| Combined physical and mental practice [59] | 1 | 1 | The location for the therapy session, the occupational therapist, patient motivation, stroke severity | Provides physical and mental practices | The patient complained of minor fatigue, increased frustration with more challenging tasks and boredom | Patient showed increases in measures of functional performance and self-perception of performance, despite persistent Ideomotor Apraxia (IMA). |
| Stroke Rehabilitation Enhancing and Guiding Transition Home (STRENGTH) [71] | 7 | 1 | N/A | Provides opportunities for therapist, client, and carers to experience the challenges of everyday activities within the home and immediate community environment | Have a better idea when participant experiencing transitions | STRENGTH allowed participants to see positive outcomes in relation to their physical, cognitive and communication abilities, promote adaption. However, due to creating false environment during STRENGTH, participant may not be ready with the dynamic situations of real environment |
| Asitaba programme [68] | 2 | 1 | Underlying problem based on technical consultation and vocational evaluation form | Supported by the work support agencies and hospital that provide evaluation. Emphasises on participant's learning ability to recognised tiredness, how it could be controlled and explain disabilities | Participant commented that his anxiety and distress toward re-employment was diminishing following the Asitaba programme. | Participant started seeking jobs by himself, which was the initial objective. Participant has re-entered the workforce and is actively working as a clerk. |

*Only adults with stroke. Other study participants such as carers and staff were not included in the total number.

** The number of young adults (18–45) included in the All age adult sample.

communication issues [38,40,44,47,55,57,63,72,81,86,87,89,101,111]. The programme outcomes resulted in improved quality of life and well-being.

Neuro-pharmacological related intervention referred to programmes that used drugs to improve neuro function, such as attention and initiation [31]. Integrated programmes refer to stroke programmes that included more than two stroke services or healthcare teams, such as hospitals, work support agencies, occupational therapist, speech pathologist, and physical therapist [59,68,71]. These types of programme mainly focused on preparing young adults with stroke return to work.

Lastly, our findings suggest that each stroke rehabilitation programme operated in a different context. The context in which each programme was embedded was likely to influence the outcome of the programme. Therefore, it is important to consider these contexts when implementing stroke rehabilitation programmes for young adults. Analysis of the societal health and welfare circumstances in which these rehabilitation programs are embedded is beyond the remit of this review.

## Young adults' experiences of stroke rehabilitation

**Acute settings.** Of the 85 included papers, 29 took place in acute settings, including inpatient and inpatient rehabilitation facilities. Twenty four of these 29 articles reported the experiences of stroke rehabilitation in the acute setting among young adults. These experiences were related to staff-patient relationships, routines and environment, and age appropriate stroke care.

*Patient-staff relationships.* The relationship between young adults with stroke and healthcare professionals has been identified as having a substantial influence on their experience as patients in acute settings [10]. Many young adults spoke of their satisfaction with their inpatient experience and highlighted how the care offered by the healthcare staff made them feel safe, comfortable, and reassured regarding their stroke recovery [52,75,77]. Trust was deemed as crucial before young adults could confide in staff [38].

Loss of independence and control were issues identified by young adults, and related to their inability to perform activities without staff assistance [66,81,102]. Therefore, empathy and emotional support from healthcare staff were highly valued by young adults [8,102]. Judgemental comments about their conditions were reported as negative and upsetting by many young adults [8,32,102]. Several also spoke of their negative experiences associated with the situations where their individual preferences were not taken into account by the healthcare staff [81,82,102].

Young adults' participation in decision making appeared to be accepted as a positive experience, particularly in discharge planning. In one case, a patient was involved in the decision making and invited to attend the multidisciplinary team meetings [53]. This involvement resulted in an agreement of living in the community after hospital discharge, whereas in the first place, the patient intended to live in his previous flat despite his disability [53]. One young patient reported that home visits a week before discharge were helpful to provide expectations of life at home and what adjustments were needed to their physical environment [71].

Effective communication between patient and staff was crucial in improving young adults' experiences in acute care. The discussion between staff and young adults about their hopes after stroke may promote optimistic thoughts [40]. The following hopes were identified: return to education, embracing spirituality and staying connected with their support system, such as friends and families [40].

Negative experiences related to a lack of communication between young adults and staff were indicated by a lack of discussion of personal goals [85], no rehabilitation assessment before hospital discharge [66], and not understanding the rehabilitation process [34]. Young adults also reported difficulty in processing information received in hospital during the acute phase of their stroke, particularly in understanding the Latin terminology and jargon used by healthcare professionals [44].

*Routines and environment.* Many young adults with stroke spoke of their routines and the environment of the inpatient rehabilitation setting. They commonly experienced boredom as a result of few activities during their hospital stay [60,81]. Participating in activities was reported as a way to occupy young adults' time, such as regular rehabilitation training [66] and attending reading sessions [38]. Rehabilitation training was favoured by young adults and the time after the exercise was considered as 'boring' [66,102]. A patient also reported feeling optimistic and hopeful after completing an eight-week physical rehabilitation programme [93]. Several patients highlighted that training intensity and duration should be increased [72,93].

With respect to the environment, staying in a single room could feel isolating: 'a prison cell' [102]. Nevertheless, sharing a room could lead to lack of privacy and the busy environment was described like 'a train station' [86]. Several young adults spoke of difficulties sleeping at the hospital due to constant noise and staff carrying out observations [8,107].

*Age-appropriate stroke care*. The provision of appropriate stroke care in terms of patient age group and conditions was central to young adults' experiences in the acute setting. Young adults often reported that they were admitted to an inpatient setting with no other patients of the same age group. This led to feelings of 'no camaraderie' and not being able to share experiences with other patients, which they felt was important [8,34,109]. Being the only young person in inpatient rehabilitation was reported to trigger feelings of terror, frustration and isolation [32] and depression [39]. Young adults reported that the inpatient facilities were designed for older people instead of young people, particularly in terms of the stroke care goals [8]. Furthermore, they mentioned that the information provided in the hospital was less relevant to young adults, for instance, the use of elderly people's pictures in the information packs and no information about returning to work, access to income support, driving post stroke and the options for contraception [8].

In contrast, positive feelings were generated from the ability to identify with people with similar conditions. Having a shared understanding with other stroke patients in the care settings was mentioned as psychologically beneficial [8]. Many young adults felt encouraged after listening and speaking to stroke survivors who had gone through situations that they could relate to [8,63].

**Community and social care settings.**   Community and social care settings (beyond inpatient) were reported in 75 out of the 85 articles in this review, with 48 reporting on the rehabilitation experiences of young adults with stroke. These findings are presented under three themes, namely transition from hospital to home, outpatient stroke rehabilitation and community rehabilitation.

*Transition from hospital to home*. The transition of young adults following stroke from inpatient to home care seemed to be a crucial point in their recovery journey. Whilst being discharged home was perceived as a sign of progress towards resuming normality and returning to personal and calm space [46,86], many spoke of the discontinuity of services after their discharge and felt they had 'slipped through the system' [8]. They felt that they had been forgotten, betrayed and had lost a sense of the safety provided by healthcare staff during their inpatient care [77]. Reported changes included struggles in carrying out activities at home previously assisted by the hospital staff [66] and the physical difference between the home and hospital environment, such as not having a non-slip floor at home [71].

Young adults often made adjustments to achieve their daily living activities and accommodate any limitations resulting from stroke. Reported strategies included prioritising essential activities, completing tasks one by one and choosing relaxing activities above house chores [9,48,97]. Another common approach was finding ways to compensate for physical impairment, such as using a trolley to carry groceries [103] and dedicating more time to complete a task [95]. Other types of adaptation such as writing lists for memory rehabilitation, were also discussed. One young adult described how doing things like writing lists felt like 'cheating' but that he was trying to not put pressure on himself [110].

*Outpatient stroke rehabilitation*. Most young adults following stroke agreed that outpatient rehabilitation helped them to progress their recovery, ranging from occupational therapy, speech and language therapy, functional electrical stimulation (FES), podiatry, to psychological therapy [45,46,65,68,87]. Positive comments were related to increased physical activity levels [47,65,83], improved speech [45], reduced anxiety from practicing relaxation techniques [87] and increased awareness of their own disability and limitation post-stroke [68]. Nevertheless, young adults reported some aspects of the rehabilitation that did not meet their expectations, including a mismatch between their own and healthcare staff's rehabilitation goal settings or no clear goal setting, a lack of information about realistic timeframes for rehabilitation, or how tiring it could be, a lack of rehabilitation activities beyond mobility function and the short

duration of each session [8,70,75]. Young adults described the importance of transparent goal setting in the context to their overall experiences and their feeling of not being in the 'driver's seat', or being understood, showing the need to address the power imbalance between patients and staff [109].

There are a number of medical devices for use in stroke rehabilitation. Young adults using SaeboFlex®; a device for upper limb rehabilitation, described how using the equipment provided them with hope despite the limited evidence on its effectiveness [106]. Other rehabilitation devices such as exoskeletons were also discussed as having limited practical function in particular whether they could be used outside and in public [108].

Occupation seems to be an important part of young adult's identity in the literature. Looking for jobs following stroke could trigger anxiety [47,68] and not being able to return to work could result in sadness and depression [47,89]. One young adult stated that returning to work was the 'best' rehabilitation and it provided a sense of returning to normality by going out and interacting with colleagues [90]. However, return-to-work or vocational rehabilitation was reported to be lacking, thus contributing to the psychological and social impact of stroke for young adults [75,105].

Young adults appeared to show determination, motivation and enthusiasm for stroke rehabilitation beyond the hospital setting. These characteristics were evidenced by high attendance rates and continuous engagement of young adults in their rehabilitation programmes [41,59,69]. Many young adults reported that they wanted to challenge themselves and hard work was required to progress [67,84,88]. A young man articulated his motivation to improve his health status and to control stroke risk factors by managing his blood pressure, whereas in the past this was not his concern [101]. Nevertheless, the repetitive nature of rehabilitation training could also lead to exasperation [89].

Many young adults spoke of their experiences of follow-up services related to information provision and content. Several mentioned receiving limited information from healthcare staff about their post-stroke condition, for instance, the physical, emotional and behavioural changes [8], or when their next brain scan would be and which professional should be contacted for advice [79]. A comprehensive list of accessible services and the information about returning to work and accessing welfare, family, or income support were desired among young adults [8,14,34,82]. They also highlighted the importance of receiving practical advice and understandable information from healthcare professionals [8,44].

Young adults appeared to experience challenges in accessing a range of stroke services in social care settings. They experienced extended delays [8,79] or no access at all to particular stroke rehabilitation services, such as psychological and community-based support [8,66]. Other young adults also mentioned having inconsistent meetings with their therapists and difficulties in reaching out to them [79]. A strict schedule of follow-up services was deemed as limiting as young adults had a range of family and work responsibilities that they needed to work around and negotiate with their spouse [79].

The relationship between young people with stroke and healthcare professionals in the context of social care settings appeared to impact on their recovery journey. Healthcare staff who provided reassurance, encouragement and motivation were reported to empower young stroke survivors in achieving higher mobilisation goals in their recovery [52,88]. In contrast, insensitive and discouraging comments from healthcare staff, for instance, stating that the patient would not be able to drive anymore, were reported to be upsetting and diminishing their hope [8].

Several young adults spoke of their plans after completing the rehabilitation programme. The reported short-term plans included returning to work as a volunteer in a day centre and moving to independent accommodation after living with her carers following stroke [67]. A

long-term goal described by a young woman was to be able to dance at her children's wedding [80]. However, cessation of rehabilitation services in the community setting seems to bring tension to young adults with stroke. Young adults expressed their concern about ending their outpatient physiotherapy sessions [52]. After the rehabilitation services were withdrawn, feelings of isolation and abandonment were commonly reported [89,94]. Nonetheless, having shared experiences with people in the community helped fill the gap that was missing from the withdrawn services [89].

*Community rehabilitation.* Many young adults reported positive benefits of participating in a community group [8,55]. Rehabilitation programmes in community settings allowed young adults to overcome and progress beyond their set self-boundaries [67]. Group interaction with other stroke survivors and learning from their successes were reported to motivate young adults to push themselves beyond their perceived limits [97].

Young adults spoke of psychological benefits of sharing similar experiences with other young people, providing peer support and encouragement, instilling hope and improving self-esteem through the socialising process [8,55]. These similar findings were also reported by aphasic young adults. They spoke of enjoying their participation in community groups, social, art and craft activities, despite having communication difficulties [76]. Other young adults with aphasia reported the benefit of volunteering in building self-confidence and independence [57].

The group composition appeared to influence young adults' rehabilitation experiences in the community context. They expressed experiencing difficulties in relating to older people with stroke due to the difference in life stage, including lifestyles and interests [72,79]. In one case of a young woman was admitted to a nursing home, she spoke of her feelings of estrangement being surrounded by older adults [18].

The delivery location of community rehabilitation seemed to be an important aspect of young adults' experiences. They mentioned that the community building used to deliver the rehabilitation service introduced them to other facilities they could access beyond the programme, for example, the gym and the pools on the same site [67]. The 'youthful' ambience of the rehabilitation centre that was described as supportive as she did not feel like a patient any more [79].

## Discussion

This review aimed to scope the experiences of young adults relating to stroke rehabilitation across the continuum of care settings. Stroke represents a major disruption to life in all dimensions (bio-psycho-social) and in particular for the young adult will impact on the individual, their family, work, activities, participation in society and lifestyle. Rehabilitation and recovery or partial recovery may take years. Such a life event forces a reappraisal and reprioritisation of life goals. In this review experiences were identified at each transition phase of the young adults' stroke journey. The journey started with the stroke and diagnosis with admission to an acute facility, followed by transfer to inpatient rehabilitation, discharge from hospital, attending rehabilitation in the community setting, and continuing to adjust to the changes resulting from stroke. We identified three key findings across these data: 1) stroke impacted young adults in both negative and positive ways, 2) the focus and the outcomes of stroke rehabilitation were predominantly physical, and 3) young adults experienced a lack of appropriate stroke rehabilitation in both acute and social care settings. These findings highlight the breadth of young adults' experiences following stroke and the support required to meet their unmet needs.

In our review, we found that young adults experienced enormous physical, financial, social and psychological impacts following stroke. The physical and psychological consequences of

stroke in young adults identified in this review are not new and are consistent with previous studies reporting the prevalence of disability, cognitive impairment and depression in stroke survivors across the age groups [112–114]. Physical deficits were perceived as the most problematic and distressing consequences of stroke that tended to lead on to psychological consequences, such as frustration and depression. The disabilities experienced by young adults with stroke were often invisible and misunderstood by others, which further intensified their frustration. With respect to financial and social aspects, the findings were consistent with a prior systematic review that working-age adults experienced financial hardships, difficulties in returning to work and challenges in maintaining relationships with family and friends, and refrained from social activities following stroke [115]. Such financial issues will be dependent on the individual and their circumstances, and the societal, health and welfare frameworks in which they live. Loss of income was a common issue and many were not able to return to work or access support funding, but relied on personal savings and financial support from family [9,35,41,64,75,82,91,95]. Hidden disabilities, such as memory, altered cognition and behavioural changes affected their abiity to return to work or gain alternative employment [82]. Financial hardship also affected their engagement with community rehabilitation opportunities [82]. Whilst the financial hardships and support required were evident in the young adults' experiences, this area requires further exploration and research.

Stroke diagnosis was reported as triggering feelings of fear and grief among young adults. Nevertheless, several young adults spoke of positive changes following their stroke, for example, that they had more time for their spouse and children and for practising their spirituality and developing their own personal strength. These negative and positive experiences suggest that the traumatic event of stroke is internalised individually. Calhoun and Tedeschi [116] refer to positive changes in response to any traumatic event as posttraumatic growth. According to the authors, there are five dimensions of posttraumatic growth, which are personal strength, new possibilities, relating to others, appreciations of life, and spiritual change. All these dimensions were found in young adults with stroke in the included literature and this finding is echoed by other studies of young survivors of natural disasters [117] and cancer [118]. Nonetheless, how these young adults with stroke obtained posttraumatic growth and the role of healthcare professionals in supporting the process remains unexplored. Future research is warranted to better understand the trajectories of posttraumatic growth among young adults following stroke over time. Understanding this could lead to the development of interventions to foster posttraumatic growth in young adults following stroke.

Across the identified stroke rehabilitation programmes for young adults, the focus and the expected outcomes were predominantly physical. Nevertheless, some programmes were limited to performing basic daily activities, such as making tea or walking. Young adults wanted higher-level physical goals beyond their ability to perform simple daily activities, for instance, preparing them to return to work. Moreover, there were different contexts or circumstances that influenced how each stroke rehabilitation programme worked (or not) for young adults, which led to different outcomes. Our findings highlight the importance of person-centred goal setting as opposed to a one-size-fits all approach. We strongly suggest that stroke rehabilitation programmes for young adults should be tailored to their personal goals, considering the different needs that young adults have from older people with stroke.

Although articles generally aimed to evaluate one outcome (e.g. physical or psychological) for each rehabilitation programme, the programmes often had impact on other aspects beyond their main aim, which was evident across the literature. For instance, Beesley et al. [55] conducted a qualitative study to evaluate the impact on wellbeing of an art therapy programme for stroke survivors. Besides the psychological impact of a feeling of accomplishment and confidence, the participants reported experiencing improvements in their speech and cognitive

function [55]. In the studies conducted by Casey et al. [69] and Valkenborghs et al. [93], the focus of rehabilitation was explicitly physical. However, the social interaction from attending such programmes had implications on psychological status and well-being of stroke survivors [69,93]. Completing the rehabilitation programme seems to be empowering for young adults, in particular by improving their future outlook enabling them to reprioritize their goals. These findings show the interconnectedness of physical, psychological and social domains in stroke rehabilitation and reinforces the need to create targeted, responsive, age appropriate rehabilitation. Further research to understand the context and the mechanisms by which stroke rehabilitation programmes succeed, or not, is needed to develop future effective interventions.

Marginalisation and invisibility were apparent in the experiences of young adults with stroke in this review, it was further reflected in comments indicating how they felt excluded from stroke rehabilitation and the healthcare system. They reported experiencing stroke-related stigmatisation by society and healthcare staff related to their stroke diagnosis at a young age. They perceived self-stigma about stroke being an older people's disease, resulting in dismissing the signs of stroke and delays in seeking medical attention. Young adults also mentioned that the stroke rehabilitation programmes were designed for older adults, and they received limited information relevant to their age group, for instance, about employment, family, and income support. It is also important to acknowledge that the healthcare settings and systems in included articles are heterogenous. Therefore, when interpreting young adults' experiences identified in our findings, these contextual differences of individual countries should be taken into account.

The review identified that young adults were only a small proportion of the total participants included in evaluation studies of stroke rehabilitation programmes. There are two plausible explanations for this phenomenon. Firstly, the authors might not focus their evaluation study on younger adults. Secondly, the available programmes related to stroke rehabilitation did not meet young adults' needs in the inpatient and social care settings, which meant that only a few of them attended the programmes. Such unmet needs were reported by the young adults within the included studies.

The lack of age-appropriate stroke care and support, particularly around hidden disabilities affecting psychosocial aspects and employment results in a mix of unmet needs for young adults, which could lead to negative experiences in their stroke recovery journey. These areas need to be investigated at the individual, family, community and societal levels to bring realistic change to the experiences of young people following stroke or brain injury.

## Limitations

It is important to note that included articles in this review were heterogeneous in terms of outcome measures, publication types and levels of reported detail. They were drawn from the literature published in English and whilst this may lead to a degree of bias towards English speaking countries, it is noted that around 98% of the published scientific literature is published in English [119]. Future action on publication policy with regard to language could address this bias and assist authors across the world to raise the profile of their research to give a more global view.

The numbers of young adults who were reported or participated in the rehabilitation programmes in each article were often small. Whilst this reflects the lower incidence of stroke amongst this group, these individuals will have many years living with the consequences of the stroke; further research focused on this younger age group is needed to equip them for their futures.

This scoping review has addressed the research questions focussing on the impact of stroke, and the rehabilitation experience and outcome for young people following stroke. It has

provided descriptions based on the available literature, but cannot be generalised due to the individual nature of the experiences and the variety of contextual issues at personal, community and societal levels.

## Recommendations

The recommendations below summarise the priorities identified above from the literature. These are grouped as Research; Service; and Society.

**Research.**

1. Further research is needed to understand the context and the mechanisms by which stroke rehabilitation programmes succeed, or not, and to develop future effective interventions.

2. Financial hardships and the support required were evident in the young adults' experiences and need further exploration and research.

3. Understanding trajectories of posttraumatic growth among young adults following stroke over time is needed for the development of interventions to foster this growth.

4. The provision of age appropriate stroke care and support needs to be investigated at the individual, family, community and societal levels. This is particularly important around hidden disabilities affecting psychosocial aspects and employment.

5. There is a need for global and country specific research on the incidence, prevalence and life expectancies of young people with stroke to understand the impact of this condition across the world.

**Service.**

1. The development of age-appropriate stroke services for young people with activities, demands and information relevant to their age group is needed, especially around employment, family, and income support.

2. Stroke rehabilitation programmes for young adults should be tailored to their personal goals, considering the different needs that young adults have from older people with stroke.

**Society.**

1. A greater awareness of stroke affecting young people is needed for both public and professionals to ensure timely recognition and action in the event of a stroke. Public engagement and knowledge exchange activities are required to widen the knowledge and awareness base.

2. Action on publication policy with regard to language which favours English-speaking communities is needed to redress the bias and assist authors across the world to raise the profile of their research for a more globally balanced view.

## Conclusion

In this review, we have gathered and summarised the experiences of young adults following stroke from literature over the past 22 years. Our findings provide insights into the negative and positive consequences of stroke on young adults' lives. The unmet needs of young adults

in terms of stroke rehabilitation and care are highlighted. This has emphasised the urgent need to research and develop effective and age-appropriate rehabilitation programmes and interventions that can support young adults following stroke.

## Supporting information

**S1 Table. Research questions, operational definitions and search terms.**
(PDF)

**S2 Table. Search strategy.**
(PDF)

**S3 Table. PRISMA-ScR checklist.**
(PDF)

## Acknowledgments

We thank Rowena Stewart for her assistance with literature searching and Chest Heart & Stroke Scotland for support in the recruitment of our patient and public involvement (PPI) group. We are grateful for insights from the members of our PPI group that shaped our understanding of young adults' rehabilitation experiences after stroke. We thank the wider group of YARNS Project partners for their support and contribution to the project.

## Author Contributions

**Conceptualization:** Colin Chandler, Catherine Clarissa, Arcellia Farosyah Putri, Lissette Aviles, HyeRi Choi, Jonathan Hewitt, Helen Noble, Joanne Reid, Aisha Holloway, Daniel Kelly.

**Data curation:** Catherine Clarissa, Arcellia Farosyah Putri, Lissette Aviles, HyeRi Choi, Emily Hennessy.

**Formal analysis:** Catherine Clarissa, Arcellia Farosyah Putri, Emily Hennessy.

**Funding acquisition:** Colin Chandler, Aisha Holloway, Daniel Kelly.

**Investigation:** Catherine Clarissa, Arcellia Farosyah Putri, Lissette Aviles, HyeRi Choi, Emily Hennessy.

**Methodology:** Colin Chandler, Catherine Clarissa, Helen Noble, Joanne Reid, Aisha Holloway, Daniel Kelly.

**Project administration:** Colin Chandler, Aisha Holloway, Daniel Kelly.

**Resources:** Colin Chandler, Catherine Clarissa, Arcellia Farosyah Putri, Lissette Aviles, HyeRi Choi.

**Supervision:** Colin Chandler, Aisha Holloway, Daniel Kelly.

**Validation:** Catherine Clarissa, Arcellia Farosyah Putri.

**Visualization:** Catherine Clarissa, Arcellia Farosyah Putri.

**Writing – original draft:** Catherine Clarissa, Arcellia Farosyah Putri.

**Writing – review & editing:** Colin Chandler, Catherine Clarissa, Arcellia Farosyah Putri, Lissette Aviles, HyeRi Choi, Jonathan Hewitt, Emily Hennessy, Helen Noble, Joanne Reid, Aisha Holloway, Daniel Kelly.

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
