## [Decision Letter · Decision Letter 0]

20 Jan 2023

PONE-D-22-31811Young Adults Rehabilitation experiences and Needs following Stroke (YARNS): A scoping review of the rehabilitation care experiences and outcomes of young adults post-strokePLOS ONE

Dear Dr. Chandler,

Thank you for submitting your manuscript to PLOS ONE. After careful consideration, we feel that it has merit but does not fully meet PLOS ONE’s publication criteria as it currently stands. Therefore, we invite you to submit a revised version of the manuscript that addresses the points raised during the review process.

We look forward to receiving your revised manuscript.

Kind regards,

Kinga Zdunek, PhD

Academic Editor

PLOS ONE

Journal Requirements:

"This work was supported by the SameYou Charity through an RCN Foundation grant."

4. One of the noted authors is a group or consortium YARNS Project Partners. In addition to naming the author group, please list the individual authors and affiliations within this group in the acknowledgments section of your manuscript. Please also indicate clearly a lead author for this group along with a contact email address.

6. We note that this manuscript is a systematic review or meta-analysis; our author guidelines therefore require that you use PRISMA guidance to help improve reporting quality of this type of study. Please upload copies of the completed PRISMA checklist as Supporting Information with a file name “PRISMA checklist”.

Reviewers' comments:

Reviewer's Responses to Questions

**Comments to the Author**

1. Is the manuscript technically sound, and do the data support the conclusions?

Reviewer #1: Yes

Reviewer #2: Partly

2. Has the statistical analysis been performed appropriately and rigorously? 

Reviewer #1: N/A

Reviewer #2: N/A

3. Have the authors made all data underlying the findings in their manuscript fully available?

Reviewer #1: Yes

Reviewer #2: Yes

4. Is the manuscript presented in an intelligible fashion and written in standard English?

Reviewer #1: Yes

Reviewer #2: Yes

5. Review Comments to the Author

Reviewer #1: I found this a very interesting, and clear paper to read. It was well structured, and informative.

The manuscript is a scoping review on the experiences of rehabilitation services for young stroke survivors. The research questions are very clear, discussing the impact of stroke on young adults, describing the focus and the expected outcomes of stroke rehabilitation; and the young adults’ experiences of stroke rehabilitation in care in different settings. The paper focuses not on the availability of rehabilitation to young people, but focuses on how they feel about it, and is a good start in identifying potential gaps that might need to be researched further. Including why most rehabilitation focuses on the elderly.

The paper concludes that this an important, and overlooked, group of individuals, particularly as the authors state that ‘almost half’ of the long-term stroke survivors are aged between 20 and 64. It In the review, the authors have interrogated a number of relevant search engines, wide range of sources. They have clearly identified and articulated an apparent gap in our knowledge.

The choice of using a scoping review rather than a systematic review is justified because of the nature of the research questions, to identify the experiences of those who have undertaken stroke rehabilitation, and to recommend further avenues for research in this area.

I have only very minor comments to make.

Minor revisions / questions

The literature search was very comprehensive, and identified a large number of results. Does this mean that the search strategy was not specific enough?

Line 103: I wonder why the authors included newspaper articles etc.? And why not assess the methodological quality of the papers? Is it because you feel that young people are missed from scientific research? Maybe a little bit more information about this choice would be interesting.

Line 150 onwards: the authors discuss the expert PPI involvement in the larger project and in this review and provided feedback on the preliminary analysis. Was this just to ensure that the analysis made sense in terms of their experiences? Or did they provide additional information on top of what was captured by the scoping review (line 154)? I’m not entirely clear.

Line 233 – The sentence about the young person inviting in the Jehovah’s witnesses reads to be a little bit judgemental. Could it be slightly rewritten? Or is there a different quotation that also demonstrates the loneliness of these individuals?

In the quotation (line 255) also person’s name appears. Does this comply with confidentiality?

Discussion

Three clear findings 1. Stroke impacted in both positive and negative ways, 2. Focus and outcomes were mostly physical 3. Young adults experienced a lack of appropriate stroke rehabilitation in both acute and social care settings. The discussion about these findings is not in this order though, it jumps from 1, to 3 then to 2 and 1 again. I found it a little confusing.

The financial hardships and the healthcare workers not recognising the needs of young people is mentioned in passing in the discussion, whereas in the results this stood out for me (around line 255). It would be nice for the discussion to give a little more depth here. Is this just the young persons’ perceptions or is this already known in the scientific literature and is an identified research area?

Line 572 onwards: The focus on young people’s experiences shows, as far as I understand it, that there is a need for improvement and targeting in the provision of rehabilitation services (reflected in their feelings of marginalisation, excluded, and experiencing delays in medical attention). I was expecting this to be a greater emphasis in the discussion and conclusion, as an avenue for future exploration. Could it be more prominently discussed?

Line: 609 onwards: The authors point out that in many cases, a focus on the physical leads onto rehabilitation in other areas, such as psychological and social. These results show that it is important and interconnected and vital for young people to be included in rehabilitation. The authors state that this means that further research is needed to understand how this happens. As you have said earlier in the paper you say that you are not assessing the rehabilitation programmes themselves perhaps clarify this by saying that it reinforces the need to create targeted rehabilitation programmes by age group, and this can be done by understanding how this happens.

Conclusion

This seems a little weak compared to the rest of the paper. The addition of a summary of the recommendations identified by the research would strengthen it. The review clearly identifies an apparent gap in our knowledge, but does it say how what they’ve learned help them to focus on how to address this gap?

Reviewer #2: Thank you for a clearly written paper on an important topic. However, from robust intentions the results are modest. It would have much more impact if you could address the following issues:

1. The practicality of using only English language publications is appreciated, but the authors do not consider the resultant bias in findings. The UK provides more than one third of studies; UK plus Australia a half. The total Spanish speaking world (in Europe and South America) does not feature, and the totality of Japan, South Korea and Taiwan as countries with advanced health systems in total yield one paper. This therefore cannot be considered a global study, or even an advanced health systems study, and should be modified accordingly.

2. The authors rightly refer to Pawson and Tilley's Context-Method-Outcome model, but give no consideration of the national health, care, and social Contexts of individual countries whose studies are cited, nor the Methods of service management and delivery therein. Thus, for instance, reference to the adverse economic impact of stroke on younger adults is only meaningful in the context of social protection and welfare payment policy, disability and unemployment support, and health and care system co-payments. In the same way that the authors indicate that patients of all ages should not be considered homogenous, so also services for stroke victims in different countries cannot be considered homogenous.

3. Tables 2 and 4 are difficult to interpret, regarding size. Did so many studies only have one young adult stroke victim participant, or are the column headings misleading? What is the practical utility of such small eligible participant studies in this analysis?

4. A simple statistical analysis of the size of the problem, such as the percentage of stroke victims who are in the younger adult category, and any trends in variation by country, would be helpful.

5. The mention at several points of suffering a stroke having positive as well as negative effects appears in lines 557-559 to be solely benefits of having more time with spouse and family, and developing spirituality. However, arguably these are not benefits of the stroke itself, but of the enforced change in lifestyle - such reprioritisation of lifestyle could be achieved by positive considerations less radical than suffering a stroke.

In summary, this paper needs a radical revision of issues including type clustering of health and care systems, contextualisation of support including financial supports and societal attitudes, and more consideration of causality of patient issues, as well as reconsideration of minimum size of eligible sample.

6. PLOS authors have the option to publish the peer review history of their article (what does this mean?). If published, this will include your full peer review and any attached files.

Reviewer #1: No

Reviewer #2: No

---

## [Author Response · Author response to Decision Letter 0]

7 Jul 2023

Response to reviewers 

(all line numbers refer to the ‘Revised Manuscript with Track Changes’) 

The response is structured using each point of the reviewers’ feedback with a response in italics below it. 

Journal Requirements: 

Please ensure that your manuscript meets PLOS ONE's style requirements, including those for file naming. The PLOS ONE style templates can be found at  

E1-1 Response: Style and file naming guidelines followed 

2. Thank you for stating in your Funding Statement:  

"This work was supported by the SameYou Charity through an RCN Foundation grant." 

Please provide an amended statement that declares *all* the funding or sources of support (whether external or internal to your organization) received during this study, as detailed online in our guide for authors at http://journals.plos.org/plosone/s/submit-now .  Please also include the statement “There was no additional external funding received for this study.” in your updated Funding Statement.  

E1-2 Response: Funding statement amended to: 

This work was funded by the RCN Foundation in partnership with SameYou. No additional external funding was received for this study. 

3. In your Data Availability statement, you have not specified where the minimal data set underlying the results described in your manuscript can be found. PLOS defines a study's minimal data set as the underlying data used to reach the conclusions drawn in the manuscript and any additional data required to replicate the reported study findings in their entirety. All PLOS journals require that the minimal data set be made fully available. For more information about our data policy, please see http://journals.plos.org/plosone/s/data-availability. 

E1-3 Response: As this is a scoping review, the relevant data are the included papers. The data availability statement is amended to: 

All relevant data are within the manuscript, the included literature are listed in Table 2 and their full references given in the reference list. 

4. One of the noted authors is a group or consortium YARNS Project Partners. In addition to naming the author group, please list the individual authors and affiliations within this group in the acknowledgments section of your manuscript. Please also indicate clearly a lead author for this group along with a contact email address. 

E1-4. The consortium has been removed from the author list and remains in the acknowledgements. 

5. Please include your full ethics statement in the ‘Methods’ section of your manuscript file. In your statement, please include the full name of the IRB or ethics committee who approved or waived your study, as well as whether or not you obtained informed written or verbal consent. If consent was waived for your study, please include this information in your statement as well.  

E1-5; The ethics statement has been added to the Methods section – lines 154,155 

6. We note that this manuscript is a systematic review or meta-analysis; our author guidelines therefore require that you use PRISMA guidance to help improve reporting quality of this type of study. Please upload copies of the completed PRISMA checklist as Supporting Information with a file name “PRISMA checklist”. 

E1-6; The manuscript is a scoping review, not a systematic review; so the PRISMA-ScR checklist has been followed. This is already included as a Supplement – S3_Table PRISMA-ScR Checklist.pdf (This file has been renamed and reuploaded) 

Reviewers' comments: 

Reviewer's Responses to Questions 

Comments to the Author 

1. Is the manuscript technically sound, and do the data support the conclusions? 

Reviewer #1: Yes 

Reviewer #2: Partly 

Shape 

2. Has the statistical analysis been performed appropriately and rigorously? 

Reviewer #1: N/A 

Reviewer #2: N/A 

Shape 

Have the authors made all data underlying the findings in their manuscript fully available? 

The PLOS Data policy requires authors to make all data underlying the findings described in their manuscript fully available without restriction, with rare exception (please refer to the Data Availability Statement in the manuscript PDF file). The data should be provided as part of the manuscript or its supporting information, or deposited to a public repository. For example, in addition to summary statistics, the data points behind means, medians and variance measures should be available. If there are restrictions on publicly sharing data—e.g. participant privacy or use of data from a third party—those must be specified. 

Reviewer #1: Yes 

Reviewer #2: Yes 

Shape 

4. Is the manuscript presented in an intelligible fashion and written in standard English? 

PLOS ONE does not copyedit accepted manuscripts, so the language in submitted articles must be clear, correct, and unambiguous. Any typographical or grammatical errors should be corrected at revision, so please note any specific errors here. 

Reviewer #1: Yes 

Reviewer #2: Yes 

Shape 

5. Review Comments to the Author 

Reviewer #1: I found this a very interesting, and clear paper to read. It was well structured, and informative. 

The manuscript is a scoping review on the experiences of rehabilitation services for young stroke survivors. The research questions are very clear, discussing the impact of stroke on young adults, describing the focus and the expected outcomes of stroke rehabilitation; and the young adults’ experiences of stroke rehabilitation in care in different settings. The paper focuses not on the availability of rehabilitation to young people, but focuses on how they feel about it, and is a good start in identifying potential gaps that might need to be researched further. Including why most rehabilitation focuses on the elderly. 

The paper concludes that this an important, and overlooked, group of individuals, particularly as the authors state that ‘almost half’ of the long-term stroke survivors are aged between 20 and 64. It In the review, the authors have interrogated a number of relevant search engines, wide range of sources. They have clearly identified and articulated an apparent gap in our knowledge. 

The choice of using a scoping review rather than a systematic review is justified because of the nature of the research questions, to identify the experiences of those who have undertaken stroke rehabilitation, and to recommend further avenues for research in this area. 

Thank you for these comments 

I have only very minor comments to make. 

Minor revisions / questions 

The literature search was very comprehensive, and identified a large number of results. Does this mean that the search strategy was not specific enough? 

R1-1 Response: The scoping review approach aims to take a broad view of the available literature; hence, a large number of results are expected. This is stated at the start of the methods section Lines 77-79, and a further explanation has been added in Lines 106-108. 

Line 103: I wonder why the authors included newspaper articles etc.? And why not assess the methodological quality of the papers? Is it because you feel that young people are missed from scientific research? Maybe a little bit more information about this choice would be interesting. 

R1-2; The scoping review approach assesses the range of available literature regardless of quality. This could have included newspaper articles but did not in this study. Whilst quality assessment is an option in this style of review, the number and range of papers made this unfeasible in this study. A sentence has been added in lines 106-108 to clarify this. 

Line 150 onwards: the authors discuss the expert PPI involvement in the larger project and in this review and provided feedback on the preliminary analysis. Was this just to ensure that the analysis made sense in terms of their experiences? Or did they provide additional information on top of what was captured by the scoping review (line 154)? I’m not entirely clear. 

R1-3; Thank you, the original text was more applicable to the overall YARNS project. This has now been amended to focus on their input to the scoping review (see lines 155-161). They did not provide additional material above what was captured in the scoping review, but provided a sense check on the preliminary analysis. 

Line 233 – The sentence about the young person inviting in the Jehovah’s witnesses reads to be a little bit judgemental. Could it be slightly rewritten? Or is there a different quotation that also demonstrates the loneliness of these individuals? 

In the quotation (line 255) also person’s name appears. Does this comply with confidentiality? 

R1-4; Thank you for spotting these errors. They have now been reworded and corrected. See lines 242-249 and line 273. 

Discussion 

Three clear findings 1. Stroke impacted in both positive and negative ways, 2. Focus and outcomes were mostly physical 3. Young adults experienced a lack of appropriate stroke rehabilitation in both acute and social care settings. The discussion about these findings is not in this order though, it jumps from 1, to 3 then to 2 and 1 again. I found it a little confusing. 

R1-5; The order of the discussion has been revised to fit better with the order of impacts. The section in lines 601-618 has been moved down to line 645-662 

The financial hardships and the healthcare workers not recognising the needs of young people is mentioned in passing in the discussion, whereas in the results this stood out for me (around line 255). It would be nice for the discussion to give a little more depth here. Is this just the young persons’ perceptions or is this already known in the scientific literature and is an identified research area? 

R1-6; The financial hardships have been revisited in the discussion with further text added – see lines 576-584 

Line 572 onwards: The focus on young people’s experiences shows, as far as I understand it, that there is a need for improvement and targeting in the provision of rehabilitation services (reflected in their feelings of marginalisation, excluded, and experiencing delays in medical attention). I was expecting this to be a greater emphasis in the discussion and conclusion, as an avenue for future exploration. Could it be more prominently discussed? 

R1-7; A paragraph has been added to the discussion to address this further (lines 663-667) 

Line: 609 onwards: The authors point out that in many cases, a focus on the physical leads onto rehabilitation in other areas, such as psychological and social. These results show that it is important and interconnected and vital for young people to be included in rehabilitation. The authors state that this means that further research is needed to understand how this happens. As you have said earlier in the paper you say that you are not assessing the rehabilitation programmes themselves perhaps clarify this by saying that it reinforces the need to create targeted rehabilitation programmes by age group, and this can be done by understanding how this happens. 

R1-8 Thank you for this comment. We have strengthened and added to the text in lines 639 -644 to address this. 

Conclusion 

This seems a little weak compared to the rest of the paper. The addition of a summary of the recommendations identified by the research would strengthen it. The review clearly identifies an apparent gap in our knowledge, but does it say how what they’ve learned help them to focus on how to address this gap? 

R1-9; Thank you, in addition to developing the discussion further as shown in the above revisions, sections on limitations and a summary of recommendations have been added – Lines 672-723. The conclusion remains a brief statement but stronger statement. Lines 726-731. 

Reviewer #2: Thank you for a clearly written paper on an important topic. However, from robust intentions the results are modest. It would have much more impact if you could address the following issues: 

1. The practicality of using only English language publications is appreciated, but the authors do not consider the resultant bias in findings. The UK provides more than one third of studies; UK plus Australia a half. The total Spanish speaking world (in Europe and South America) does not feature, and the totality of Japan, South Korea and Taiwan as countries with advanced health systems in total yield one paper. This therefore cannot be considered a global study, or even an advanced health systems study, and should be modified accordingly. 

R2-1; Thank you for this comment. It was indeed only practical for us to search the literature published in English. This is a limitation noted in lines 674-679. 

Whilst this is a scoping review aiming to take a wide view of the available literature, it does not claim to be global. This has been clarified in edits in Lines 33,34; and is consistent with the text in lines 185-187 and summarised in Table 3. 

this point is also picked up in the recommendations summary lines 721-723. 

2. The authors rightly refer to Pawson and Tilley's Context-Method-Outcome model, but give no consideration of the national health, care, and social Contexts of individual countries whose studies are cited, nor the Methods of service management and delivery therein. Thus, for instance, reference to the adverse economic impact of stroke on younger adults is only meaningful in the context of social protection and welfare payment policy, disability and unemployment support, and health and care system co-payments. In the same way that the authors indicate that patients of all ages should not be considered homogenous, so also services for stroke victims in different countries cannot be considered homogenous. 

R2-2; A good point, but this is outside the remit of this scoping review which is focussed on the experiences of young adults post-stroke Lines 313, 350-352. 

It is picked up in the Limitations section Lines 685-690; 

3. Tables 2 and 4 are difficult to interpret, regarding size. Did so many studies only have one young adult stroke victim participant, or are the column head

---

## [Decision Letter · Decision Letter 1]

23 Jan 2024

PONE-D-22-31811R1Young Adults Rehabilitation experiences and Needs following Stroke (YARNS): A scoping review of the rehabilitation care experiences and outcomes of young adults post-strokePLOS ONE

Dear Dr. Chandler,

Thank you for submitting your manuscript to PLOS ONE. After careful consideration, we feel that it has merit but does not fully meet PLOS ONE’s publication criteria as it currently stands. Therefore, we invite you to submit a revised version of the manuscript that addresses the points raised during the review process.

We look forward to receiving your revised manuscript.

Kind regards,

Nadinne Alexandra Roman, Ph.D.

Academic Editor

PLOS ONE

Journal Requirements:

Reviewers' comments:

Reviewer's Responses to Questions

**Comments to the Author**

1. If the authors have adequately addressed your comments raised in a previous round of review and you feel that this manuscript is now acceptable for publication, you may indicate that here to bypass the “Comments to the Author” section, enter your conflict of interest statement in the “Confidential to Editor” section, and submit your "Accept" recommendation.

Reviewer #2: (No Response)

Reviewer #3: All comments have been addressed

2. Is the manuscript technically sound, and do the data support the conclusions?

Reviewer #2: Yes

Reviewer #3: Yes

3. Has the statistical analysis been performed appropriately and rigorously? 

Reviewer #2: (No Response)

Reviewer #3: Yes

4. Have the authors made all data underlying the findings in their manuscript fully available?

Reviewer #2: Yes

Reviewer #3: Yes

5. Is the manuscript presented in an intelligible fashion and written in standard English?

Reviewer #2: Yes

Reviewer #3: Yes

6. Review Comments to the Author

Reviewer #2: Thank you for submitting an improved manuscript. There are just a few points for consideration:

1. The sentence on lines 41-42 puts positive impact before negative, whereas the greatest weight of impact, and the focus of rehabilitation, is rightly on the negative effects, and this is the focus of the paper. The Impact section on lines 191 et seq is much more appropriately balanced. Thus the sentence on lines 41-42 needs removing, and the point about some positive impacts put as a clause towards the end of the Abstract's Results section.

2. Line 181 - the comma is misplaced, and should appear after "literature".

3. Line 241 - the dash is inappropriate and should be removed.

4. "wolf" should be "Wolf" (author's name).

5. I still consider that in the Discussion there should be some recognition that health, care, and social welfare systems are not universally homogenous across the globe, and their nature may have influence on the outcome of service provision in individual countries or settings.

Reviewer #3: Thank you for the opportunity to read an interesting paper on Stroke. The paper is written clearly with lots of information to read about.

1. The research question is clear and appropriate for the scoping review.

2. The abstract delivers adequate information for a reader to understand the entire paper.

3. Introduction: Well written and delivers adequate overview of the condition.

4. Methods:

I have only minor comments to address:

The review was structured on Arksey and O'Malley methodological framework for scoping studies. The framework has six stages: The paragraphs that are written under the methodology can be fit under the six stages listed below which makes a reader to better understand the steps that were followed. For example: 1. Identifying the research question—the paragraph can have a heading---similar to follow for the others.

Name of the databases can be elaborated at the first use CINAHL (Cumulative Index to Nursing and Allied Health Literature)

The authors describe as All types of publications in journal articles---could it be mentioned as original studies or reviews.

5. Results:

Line 184, the authors describe as majority of studies could it be replaced as most of the studies?

6. Discussion is well -written and it justifies the purpose of this review.

7. PLOS authors have the option to publish the peer review history of their article (what does this mean?). If published, this will include your full peer review and any attached files.

Reviewer #2: No

Reviewer #3: No

---

## [Author Response · Author response to Decision Letter 1]

6 Mar 2024

We thank the editor and reviewer for their comments and suggested revisions. All suggestions have been addressed. Detail of these revisions are given in the file Response to Reviewers.

---

## [Decision Letter · Decision Letter 2]

17 Nov 2024

PONE-D-22-31811R2Young Adults Rehabilitation experiences and Needs following Stroke (YARNS): A scoping review of the rehabilitation care experiences and outcomes of young adults post-strokePLOS ONE

Dear Dr. Chandler,

Thank you for submitting your manuscript to PLOS ONE. After careful consideration, we feel that it has merit but does not fully meet PLOS ONE’s publication criteria as it currently stands. Therefore, we invite you to submit a revised version of the manuscript that addresses the points raised during the review process.

We look forward to receiving your revised manuscript.

Kind regards,

Diphale Joyce Mothabeng, PhD

Academic Editor

PLOS ONE

Journal Requirements:

Additional Editor Comments:

Thank you for your submission. The journal will be in contact with you regarding the next steps.

Reviewers' comments:

Reviewer's Responses to Questions

**Comments to the Author**

1. If the authors have adequately addressed your comments raised in a previous round of review and you feel that this manuscript is now acceptable for publication, you may indicate that here to bypass the “Comments to the Author” section, enter your conflict of interest statement in the “Confidential to Editor” section, and submit your "Accept" recommendation.

Reviewer #2: All comments have been addressed

Reviewer #4: (No Response)

2. Is the manuscript technically sound, and do the data support the conclusions?

Reviewer #2: Yes

Reviewer #4: Yes

3. Has the statistical analysis been performed appropriately and rigorously? 

Reviewer #2: N/A

Reviewer #4: Yes

4. Have the authors made all data underlying the findings in their manuscript fully available?

Reviewer #2: Yes

Reviewer #4: Yes

5. Is the manuscript presented in an intelligible fashion and written in standard English?

Reviewer #2: Yes

Reviewer #4: Yes

6. Review Comments to the Author

Reviewer #2: Thank you for fully implementing the previous feedback. This thoroughness is appreciated and improves a good manuscript.

Reviewer #4: Overview

First, I commend the authors for the efforts and rigour invested into this study. This scoping review sought to explore young adults rehabilitation experiences and needs following stroke (YARNS): A scoping review of the rehabilitation care experiences and outcomes of young adults post-stroke. According to the authors,” results highlight the unmet needs of young adults in their stroke recovery journey”. Authors concluded thus “Effective rehabilitation programmes and interventions should be developed to support young adults following stroke and meet their age-specific needs”

Areas of improvements

This is elegant piece of literature. The methodology was sound and the theoretical framework applied throughout the study. Also, the language is standard. However, a few minor concerns have been highlighted in the manuscript text.

Decision

I recommend the publication of the manuscript following minor revision.

7. PLOS authors have the option to publish the peer review history of their article (what does this mean?). If published, this will include your full peer review and any attached files.

Reviewer #2: No

Reviewer #4: No

---

## [Author Response · Author response to Decision Letter 2]

18 Dec 2024

Reference list has been reviewed and two minor updates made.

Reviewer 4 comments on the manuscript have been addressed, even though it appears he reviewed an outdated version of the manuscript (not the March 2024 submission)

Details of these are given in the response to reviewer document

---

## [Decision Letter · Decision Letter 3]

10 Jan 2025

Young Adults Rehabilitation experiences and Needs following Stroke (YARNS): A scoping review of the rehabilitation care experiences and outcomes of young adults post-stroke

PONE-D-22-31811R3

Dear Dr. Chandler,

We’re pleased to inform you that your manuscript has been judged scientifically suitable for publication and will be formally accepted for publication once it meets all outstanding technical requirements.

Kind regards,

I Gede Juanamasta

Academic Editor

PLOS ONE

Additional Editor Comments (optional):

Reviewers' comments:

Reviewer's Responses to Questions

**Comments to the Author**

1. If the authors have adequately addressed your comments raised in a previous round of review and you feel that this manuscript is now acceptable for publication, you may indicate that here to bypass the “Comments to the Author” section, enter your conflict of interest statement in the “Confidential to Editor” section, and submit your "Accept" recommendation.

Reviewer #2: All comments have been addressed

Reviewer #5: All comments have been addressed

2. Is the manuscript technically sound, and do the data support the conclusions?

Reviewer #2: Yes

Reviewer #5: Yes

3. Has the statistical analysis been performed appropriately and rigorously? 

Reviewer #2: Yes

Reviewer #5: Yes

4. Have the authors made all data underlying the findings in their manuscript fully available?

Reviewer #2: Yes

Reviewer #5: Yes

5. Is the manuscript presented in an intelligible fashion and written in standard English?

Reviewer #2: Yes

Reviewer #5: Yes

6. Review Comments to the Author

Reviewer #2: (No Response)

Reviewer #5: (No Response)

7. PLOS authors have the option to publish the peer review history of their article (what does this mean?). If published, this will include your full peer review and any attached files.

Reviewer #2: No

Reviewer #5: No

---

## [Editor Report · Acceptance letter]

14 Jan 2025

PONE-D-22-31811R3 

PLOS ONE

Dear Dr. Chandler, 

I'm pleased to inform you that your manuscript has been deemed suitable for publication in PLOS ONE. Congratulations! Your manuscript is now being handed over to our production team.

Kind regards, 

on behalf of

Dr. I Gede Juanamasta 

Academic Editor

PLOS ONE
